# Altered brain responses to noxious dentoalveolar stimuli in high-impact temporomandibular disorder pain patients

Connor M. Peck[ID][1], David A. Bereiter[1], Lynn E. Eberly[2], Christophe Lenglet[3], Estephan J. Moana-Filho[ID][1]*

1 Department of Diagnostic and Biological Sciences, University of Minnesota School of Dentistry, Minneapolis, Minnesota, United States of America, 2 Division of Biostatistics, University of Minnesota School of Public Health, Minneapolis, Minnesota, United States of America, 3 Department of Radiology, University of Minnesota Medical School, Minneapolis, Minnesota, United States of America

* moana004@umn.edu

**Data Availability Statement:** The data set for this article has been published via Harvard Dataverse, DOI: doi.org/10.7910/DVN/BUBB5I.

## Abstract

High-impact temporomandibular disorder (TMD) pain may involve brain mechanisms related to maladaptive central pain modulation. We investigated brain responses to stimulation of trigeminal sites not typically associated with TMD pain by applying noxious dentoalveolar pressure to high- and low-impact TMD pain cases and pain-free controls during functional magnetic resonance imaging (fMRI). Fifty female participants were recruited and assigned to one of three groups based on the Diagnostic Criteria for Temporomandibular Disorders (DC/TMD) and Graded Chronic Pain Scale: controls (n = 17), low-impact (n = 17) and high-impact TMD (n = 16). Multimodal whole-brain MRI was acquired following the Human Connectome Project Lifespan protocol, including stimulus-evoked fMRI scans during which painful dentoalveolar pressure was applied to the buccal gingiva of participants. Group analyses were performed using non-parametric permutation tests for parcellated cortical and subcortical neuroimaging data. There were no significant between-group differences for brain activations/deactivations evoked by the noxious dentoalveolar pressure. For individual group mean activations/deactivations, a gradient in the number of parcels surviving thresholding was found according to the TMD pain grade, with the highest number seen in the high-impact group. Among the brain regions activated in chronic TMD pain groups were those previously implicated in sensory-discriminative and motivational-affective pain processing. These results suggest that dentoalveolar pressure pain evokes abnormal brain responses to sensory processing of noxious stimuli in high-impact TMD pain participants, which supports the presence of maladaptive brain plasticity in chronic TMD pain.

## Introduction

Temporomandibular disorder (TMD) is a heterogenous group of conditions affecting the masticatory muscles and/or temporomandibular joints (TMJ) [1]. The main clinical characteristics associated with TMD include limited mandibular range of motion, TMJ sounds associated

**Funding:** Research reported in this publication was supported by the National Institute of Dental & Craniofacial Research of the National Institutes of Health (https://www.nidcr.nih.gov) under Award Number R00 DE027414 (E.J.M.). The content is solely the responsibility of the authors and does not necessarily represent the official views of the National Institutes of Health. Additional support was provided by the Office of the Vice President for Research, University of Minnesota. The funders had no role in study design, data collection and analysis, decision to publish, or preparation of the manuscript.

**Competing interests:** The authors have declared that no competing interests exist.

with jaw functioning, and pain over the pre-auricular, cheek, and/or temporal areas [2]. TMD is the most common chronic orofacial pain condition [3], affecting around 5% to 12% of the population with an estimated annual cost of $4 billion [4]. About half to two-thirds of TMD patients seek treatment, of whom approximately 15% develop chronic TMD [4]. Recent research surrounding painful TMD has dichotomized TMD pain-related disability as pain impact status (low, high) using the Graded Chronic Pain Scale (GCPS) [5,6]. Patients with high-impact TMD pain experience greater jaw functional limitation and psychological unease [7] while incurring increased health care costs over a 6-month period of roughly $525 more than low-impact TMD pain patients [8].

Chronic TMD pain usually manifests clinically as myalgia affecting the masticatory muscles, with or without concurrent TMJ arthralgia [9]. While acute myalgia is generally believed to involve peripheral sensitization attributed to tissue trauma, mechanical overloading, neurogenic inflammation, and ischemia [10], chronic myogenous TMD pain is often associated with maladaptive central nervous system (CNS) processes, resulting in pain perception that persists even without apparent peripheral nociceptive drive [11,12]. Those CNS processes are collectively called central sensitization [13,14], defined as an increased responsiveness of CNS nociceptive neurons to normal or subthreshold afferent input [15]. The term maladaptive central pain modulation (MCPM) may be more appropriate in encompassing the various central processes that can be dysfunctional in chronic pain states in addition to central sensitization. In chronic TMD pain patients it is well accepted that MCPM plays an important role in the manifestation of clinical symptoms, an assumption that is supported by findings of hyperalgesia [16], increased endogenous pain facilitation and impaired inhibition [17] in this patient population.

Neuroimaging studies have further implicated CNS dysregulation in chronic TMD pain, showing that patients exhibit structural [18–21], functional [22–29], and neurochemical [28,30,31] alterations involving brain regions associated with pain processing [12,32]. Stimulus-evoked functional magnetic resonance imaging (stimfMRI) is commonly used to investigate CNS mechanisms underlying pain processing in humans [32]. In chronic TMD pain patients, stimfMRI studies have applied somatosensory stimuli to trigeminal sites typically reported as painful in chronic TMD, such as the temporalis muscle [22,33], but also to non-painful distant sites [24,25,28,33–35]. Thus, the latter studies stimulated sites that should not be influenced by peripheral sensitization of TMD-affected body structures, nonetheless these studies reported alterations in the insula [25,34], primary (SI) [25,34] and secondary (SII) somatosensory cortices [25], anterior cingulate cortex (ACC) [25,33], thalamus [25], dorsolateral prefrontal cortex [24], and premotor cortex [24] in chronic TMD patients relative to controls. These results suggest widespread abnormal modulation of somatosensory processing by higher-order CNS regions in chronic TMD pain.

Dysfunction may also occur in lower regions of the CNS such as in the spinal trigeminal nucleus in the brainstem, the termination site for primary afferent fibers with receptive fields over the TMJs and masticatory muscles [36]. This likely contributes to the manifestation of MCPM and consequent referred pain, characteristics associated with chronic TMD pain [13,14,37,38]. These potential brain mechanisms offer theoretical support to the notion that noxious stimuli applied to non-painful sites in chronic TMD pain patients may evoke aberrant brain activations reflective of maladaptive CNS plasticity. However, few stimfMRI studies have applied noxious stimuli to body sites unassociated with TMD pain in both patients and controls. One study revealed decreased brain activations in regions of the frontoparietal attention network in TMD patients subjected to forearm heat [24] while another found no brain activation differences between TMD patients and controls in response to thumbnail pressure pain [33]. In a previous stimfMRI study done by our group, noxious dentoalveolar pressure stimuli

was applied to patients with persistent dentoalveolar pain disorder (PDAP), a chronic orofacial pain condition affecting the dentoalveolar structures [39] which is commonly comorbid with TMD pain [40]. Compared to controls, PDAP patients exhibited a greater extent of brain activations over several brain areas including SI, SII, insula, prefrontal cortex, and thalamus. Given that 50% of PDAP patients are also diagnosed with TMD [40,41], these findings raise the possibility that chronic TMD pain patients may also present abnormally amplified brain responses following noxious dentoalveolar pressure stimulation, in a similar fashion to PDAP patients, even though this is not a site commonly reported as painful by that patient population.

Our primary aim was to determine whether brain activations in response to noxious dentoalveolar pressure stimuli in chronic TMD pain patients differ from those in pain-free controls. We also aimed to investigate the CNS effects of pain impact status related to chronic TMD pain, hypothesizing that high-impact TMD pain patients would present with greater brain activations in regions typically associated with the processing of noxious dentoalveolar stimuli.

## Methods

The present cross-sectional study is an extension of a parent research project detailed previously [42]. The protocol was approved by the University of Minnesota Institutional Review Board (IRB). Study participation involved three experimental visits separated by 2–7 days. The first two visits took place at the University of Minnesota School of Dentistry (UMN SOD) Oral Health Clinical Research Center (OHCRC) and the third visit was held at the Center for Magnetic Resonance Research (CMRR).

### Participants

Potential participants were recruited from several sources between March 2016 and February 2020, including the patient population from the UMN TMD, Orofacial Pain and Dental Sleep Medicine Clinic (UMN TMD Clinic), flyers advertising the study on campus, and various online resources. Potential participants were screened in person at the UMN TMD Clinic or via telephone using an IRB-approved script to assess eligibility based on the study's inclusion/exclusion criteria. Only female participants were included as a measure to reduce sample heterogeneity and also based on the evidence that chronic TMD pain is at least three times more common in females than males [43].

Participants were divided in three groups: a pain-free control group and two groups of chronic TMD pain cases classified based on their pain-impact status as determined by the GCPS dichotomization. The GCPS is a validated instrument used in the Diagnostic Criteria for TMD (DC/TMD) Axis II that considers the patient's level of psychosocial adaptation to TMD pain as a guide to treatment [44,45]. It uses a hierarchical scale with grades ranging from I to IV (grade 0 means pain-free) indicating escalation of TMD pain-related disability. The chronic TMD pain groups in the present study were classified as low-impact (grades I-IIa) and high-impact (grades IIb-IV) TMD pain as done previously by others [45].

Inclusion criteria for chronic TMD pain participants consisted of 1) Adults (18 years or older); 2) Fulfillment of the validated DC/TMD criteria for myalgia with or without concurrent arthralgia and/or headache associated with TMD; and 3) Pain present for a minimum of 6 months and ≥15 days of pain in the previous 30 days. The threshold of 6 months for chronic TMD classification was adopted from criteria used by the Orofacial Pain: Prospective Evaluation and Risk Assessment (OPPERA) study [43]. Inclusion criteria for controls were defined as adults not fulfilling criteria #2 and 3 listed above.

Exclusion criteria for all participants have been detailed previously [42]. Briefly, these criteria included the following as assessed by self-report: current pain medication use that cannot be stopped <1 day prior to testing; conditions associated with altered pain perception; systemic arthritic disease, vascular disorders, neurological disorders, or neoplasia; substance abuse; MRI contraindications; adults lacking capacity to provide informed consent; and non-English speakers.

## Experimental protocol

**Informed consent, TMD examination & questionnaires (Visit 1).** Participants provided written and verbal informed consent after a detailed review of the experimental protocol. A standardized TMD examination was performed by a trained examiner according to the DC/TMD protocol. Sociodemographic, anthropometric, clinical pain, and psychosocial characteristics were collected during this visit as outlined previously [42] and are briefly described below.

Clinical pain data collected included the following: a comorbidity index assessing the self-reported presence of 18 comorbid conditions; duration of jaw pain since onset; laterality of jaw pain; jaw pain intensity using a 0 to 100 numerical rating scale (NRS); ratings of current orofacial pain intensity and unpleasantness using Gracely Box Scales [46]; total number of body sites (0–45) where significant pain was felt in the past 30 days using a pain drawing [44]; and characteristic pain intensity (CPI; 0–100) from the GCPS.

Confounding factors for pain sensitivity were recorded, including the time of day of the appointment, menstrual cycle phase information, and use of medications and caffeine in the 24 hours preceding the visit. Participants were asked to avoid acute pain medications (e.g., acetaminophen, ibuprofen, opioids) within 24 hours of each study visit start time. The Medication Quantification Scale (MQS) was used to summarize the use of all other pain medications in the last 24 hours [47].

In addition to the GCPS, psychosocial questionnaires from the DC/TMD Axis II [44] were completed to assess the following measures: jaw function specific to TMD using the Jaw Function Limitation Scale-long form (JFLS-20); depression using the Patient Health Questionnaire-9 (PHQ-9); anxiety using the Generalized Anxiety Disorder-7 (GAD-7); overall somatic symptom severity using Patient Health Questionnaire-15 (PHQ-15); and maladaptive oral parafunction using the Oral Behavior Checklist (OBC). Additionally, the Perceived Stress Scale (PSS) [48], Pittsburgh Sleep Quality Index (PSQI) [49], and Edinburgh Handedness Inventory (EHI) [50] were used to assess perceived stress, sleep quality, and handedness, respectively. Participants completed the DC/TMD Patient History Questionnaire which assessed characteristics related to jaw pain, headaches, and sociodemographic information. The order of these forms was randomized for each participant to reduce the effects of order bias and respondent fatigue. All forms were inspected for completeness to prevent missing data.

**Intraoral stimulus device (Visit 2).** Study forms were completed to update clinical pain characteristics (current pain intensity using 0–100 NRS, jaw pain location, Gracely Box Scales) and pain sensitivity confounders (time of day, date of last period, caffeine and medication use in past 24 hours).

For purposes of the present study, the primary goal of visit two was to customize the dentoalveolar stimulus device and determine the individual dentoalveolar pain threshold. The validated, MRI-compatible dentoalveolar stimulus device used to deliver pressure-pain stimuli to the gingiva has been described previously [39,51]. It was manually controlled by the examiner and used to deliver a range of mechanical pressure stimuli to anterior and posterior buccal gingival surfaces in all four intraoral quadrants.

An impression of the participant's bite was recorded on a bite bar using a vinyl polysiloxane automix system (3M ESPE Express STD VPS). The customized bite bar was subsequently mounted on the stimulus device and positioned in the participant's mouth. The stimulus location was preselected for each participant to ensure that stimulus locations were matched across groups. A range of one to eight 1/8" elastic bands were used in the device to deliver enough mechanical pressure to dentoalveolar tissues to be rated within 3-5/10 NRS pain intensity. The source of pressure was an intraoral plastic probe with a contacting surface of approximately 2 mm$^2$. The probe was positioned about 1 mm away from the buccal gingiva in the premolar region of the selected quadrant, so no pain was elicited with the probe in an "OFF" position. To deliver the stimuli, the examiner rotated a knob connected to the device forward and back at approximately 1 Hz frequency, resulting in antero-posterior probe oscillations of approximately 5 to 7 mm over the dentoalveolar process.

Once the dentoalveolar pressure pain threshold was determined, three pairs of 30 seconds ON and OFF stimulus blocks were administered. During OFF blocks, the probe did not contact the gingiva. Participants provided 0–10 NRS ratings with their hands following each ON block. Based on these ratings, the device was adjusted as necessary until consistent target pain ratings of 3-5/10 were acquired or the maximum pressure was reached (8 x 1/8" elastic bands). The device was then removed from the participants' mouth and disinfected, then placed in a labeled plastic bag and securely stored until the neuroimaging visit.

**Neuroimaging session (Visit 3).** Prior to entering the scanner room MRI safety information was reviewed and the participant completed the study forms to update clinical pain characteristics and pain sensitivity confounders. The participant prepared for the MRI session by removing all magnetic non-safe objects and changing into scrubs. Once in the scanner room, ear plugs were provided for noise protection and a Vitamin E capsule was taped to the right temple as a laterality marker. The participant's head was stabilized with non-magnetic foam cushions to prevent motion during the scan. An emergency button was placed in the patient's hand which triggered a buzzer to stop the scan if required.

A Siemens 3-Tesla Prisma MRI scanner with a 32-channel, circularly polarized radiofrequency transmit/receive head coil was used. Multimodal, whole-brain MRI data acquisition was based on the Human Connectome Project (HCP) Lifespan protocol [52]. The total scanning time was approximately 90 minutes, including a short break for the stimulus device placement as described below. The present study focused on four imaging modalities collected: two resting state fMRI (restfMRI) scans, two high-resolution T1-weighted (T1w) scans, one high-resolution T2-weighted (T2w) scan, and two stimulus-evoked fMRI (stimfMRI) scans. Four diffusion-weighted scans were also acquired but were not included in the present analyses.

Structural neuroimaging data acquisition began with T1w 3D magnetization prepared rapid gradient echo (MPRAGE) anatomical images (TR = 2400 ms; TE = 2.22 ms; voxel size = 0.8 mm$^3$ isotropic; slices per slab = 208; FoV read = 256 mm; Filter: Prescan Normalize; TI = 1000 ms; FA = 8˚; PAT mode = GRAPPA; Accel. Factor PE = 2; Multi-slice mode: Single shot; Fat suppr.: Water excit. Fast; acquisition time per run = 6m38s). Next, T2w 3D sampling perfection with application-optimized contrasts by using flip angle evolution (SPACE) images were acquired (TR = 3200 ms; TE = 563 ms; FA = NA; voxel size = 0.8 mm$^3$ isotropic; FoV read = 256 mm; Filter: Prescan Normalize; TI = NA; PAT mode = GRAPPA; Accel. factor PE = 2; Fat suppr.: None; acquisition time per run = 5m57s).

Prior to the acquisition of each pair of restfMRI and stimfMRI scans, a pair of spin echo field maps were collected using reversed phase encoding directions (AP, PA). Participants were instructed to keep their eyes open during both restfMRI scans, which were acquired using T2*-weighted blood-oxygenation-level dependent (BOLD) echo-planar imaging (EPI), each with reversed phase encoding directions (AP, PA) (TR = 937 ms; TE = 37 ms; voxel

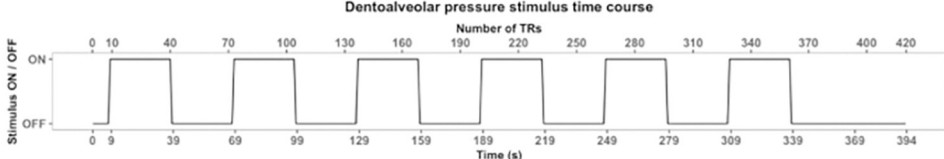

**Fig 1. Dentoalveolar pressure stimulus time course.** At the 9 second mark, the examiner adjacent to the scanner bore began applying the dentoalveolar pressure pain stimulus at a 1 Hz frequency for 30 seconds (ON block), followed by alternating 30 second OFF/ON blocks as illustrated until the completion of each scan. TR = repetition time.

size = 2.0 mm³ isotropic; FoV read = 208 mm; Multi-band accel. factor = 8; Filter: None; FA = 52˚; Delay in TR = 0 ms; PAT mode = None; Multi-slice mode: Interleaved; Fat suppr.: Fat sat.; Echo spacing = 0.58 ms; acquisition time per run = 6m45s; 420 time points).

Prior to the stimfMRI scans, the participant was withdrawn from the scanner bore for fitting of the dentoalveolar stimulus device. The examiner entered the scanner room and positioned the pre-calibrated dentoalveolar stimulus device in the participant's mouth and secured it to the head coil with tape. A support arch holding the knob used by the examiner to deliver the stimulus was secured over the scanner bed near the level of the participant's knees. A long non-magnetic stick attached to this knob was connected to the stimulus device on the opposite end so that a 90˚ knob rotation elicited full intraoral probe movement [51]. Prior to re-entering the scanner bore, the examiner tested the stimulus device and asked the participant to provide a pain rating (0–10 NRS) for a brief stimulation of approximately 30 s. If the pain rating was outside of the 3-5/10 NRS target intensity, the device was adjusted until the target pain rating was met.

Once the participant was repositioned at the scanner isocenter, a second T1w image was collected. At this point, the examiner entered the scanner room, using ear protection, and stood adjacent to the scanner bed in order to deliver the stimuli during the stimfMRI acquisition. Another set of spin-echo field maps (PA, AP) was acquired followed by the acquisition of two stimfMRI BOLD runs (AP, PA) for each participant with the same imaging parameters as the restfMRI scans outlined earlier. During these scans, the examiner applied the dentoalveolar pressure pain stimulus by rotating the knob 90˚ back and forth at a 1 Hz frequency in a blocked design fashion as outlined in Fig 1 (9 seconds initial baseline (OFF), followed by 6 alternating ON/OFF blocks of 30 seconds duration).

In order to time sync stimuli and stimfMRI data acquisition, the MRI technologist operating the scanner visually signaled to the examiner when the scan began, cuing the examiner to start the chronometer secured around their neck to accurately time stimuli delivery. Once each stimfMRI run was completed, the participant signaled the overall pain evoked by the stimuli to the examiner by using their right hand (0–5 fingers, twice if pain rating > 5). The examiner then relayed this information to the MR technologist who documented the rating on the pertinent study form. Following the completion of the second stimfMRI scan, the participant was immediately withdrawn from the scanner and the stimulus device removed. Prior to dismissal, the examiner performed a brief intraoral examination to inspect for any tissue damage caused by the stimuli, and if present the participant was instructed on how to protect the affected area until healed.

## Data analysis

All data entries recorded on paper forms during the study visits were manually transferred into an electronic database by two members of the study team (CP, EJM). These two data sets

were compared using an Excel spreadsheet which alerted the study team of any discrepancies. Two examiners (CP, EJM) reconciled any discrepancies through consensus following review of the original data forms.

Descriptive statistics were generated using the R software version 4.0.2 (R Foundation for Statistical Computing, Vienna, Austria). Data characteristics were assessed through visualization of the data plots and summary statistics tables. Statistical tests for between-group differences for anthropometric, psychosocial, and pain measures were done using jamovi version 1.2.27.0 (The jamovi project (2020), Retrieved from https://www.jamovi.org). Data was checked for normality using the Shapiro-Wilk test. Games-Howell post-hoc tests for pairwise comparisons were performed to determine whether between-group mean differences were significant. To test for mean differences between only two groups, the independent samples $t$ test assuming equal variances was used. For data not meeting parametric assumptions, between-group differences were analyzed using the Mann-Whitney $U$ test. Fisher's exact test was used to assess for differences between categorical variables. Results were reported as mean ± SD and the significance threshold was set at $p = 0.05$.

### Neuroimaging data processing

The neuroimaging data were processed using the UMN Minnesota Supercomputing Institute (MSI) computer systems. The initial step was the conversion of raw DICOM files into NIFTI format using dcm2niix [53], followed by preprocessing steps using a minimally modified version of the HCP preprocessing pipelines (v4.2.0) [54]. stimfMRI level 1 (individual stimfMRI run) and level 2 (within-participant averaged stimfMRI runs) processing was done using Task fMRI HCP scripts <https://github.com/Washington-University/HCPpipelines>. Finally, stimfMRI group analyses were performed using non-parametric permutation methods for dense and parcellated data. Main software suites used for neuroimaging data processing were FSL (v6.0.2) [55], Freesurfer (v6.0) [56] and Connectome Workbench (v1.4.2) <https://github.com/Washington-University/workbench>.

**Minimal preprocessing using HCP pipelines.** Following DICOM to NIFTI file conversion, the HCP minimal preprocessing pipelines [54] were used with minor modifications to preprocess the neuroimaging data. Briefly, structural preprocessing steps (PreFreeSurfer, Freesurfer, PostFreeSurfer) were implemented using the Montréal Neurological Institute (MNI) space templates at 0.8 mm$^3$ resolution (matching the resolution for the T1w and T2w data acquired) and gradient distortion correction using a scanner-specific gradient coefficients file. Functional preprocessing steps (fMRIVolume, fMRISurface) also included gradient distortion correction, with no additional smoothing done so that the final fMRI data resolution was 2 mm$^3$. Output files were in CIFTI format, which includes 91,282 grayordinates across three components: two for the cortical surfaces (left, right) and one for subcortical voxels.

Additional processing of functional data included: 1. Motion regression and artifact removal by feeding the fMRI BOLD data in its entirety (resting state and stimulus-evoked) to the ICA (independent component analysis, [57]) and FIX (FMRIB's ICA-based X-noiseifier, [58,59]) denoising process (ICA-FIX); and 2. Surface-based areal alignment using a joint multimodal surface matching (MSMAll) algorithm to register each participant's ICA-FIX denoised fMRI time series cortical data to a group template for improved intersubject registration of functional cortical areas [60,61].

Quality control for T1w, T2w, and BOLD fMRI raw NIFTI data files was done visually by two reviewers (CP, EJM) using the MRIQC tool [62]. If needed, raw images were inspected further using FSLeyes and a third reviewer (CL) was available to settle disagreements. Additional quality control checks were performed following completion of each major step of the

HCP pipelines and functional data processing, and any issues detected were remediated as needed (e.g., skull stripping errors during structural preprocessing).

**stimfMRI data analysis and statistics.** General linear model (GLM) analyses for ICA-FIX denoised, MSMAll surface-registered stimfMRI data were run using Task fMRI scripts distributed with the HCP pipelines. These analyses were done in two ways: 1. Dense, which includes all 91,282 grayordinates for the cortical and subcortical components as data points; and 2. Parcellated, where the grayordinate cortical data is averaged within neurobiologically meaningful parcels, and subcortical data is averaged across known subcortical structures with boundaries determined from functional connectivity as described below.

A modified version of the HCP's multi-modal parcellation version 1.0 (HCP_MMP1.0) [61] was developed for this study (Fig 2). The modification implemented was the segmentation of primary somatosensory cortex (SI) parcels into somatotopic subregions (face, upper limb, eye, trunk, lower limb), based on boundaries defined from resting state functional connectivity gradients and somatotopic task fMRI contrasts from the publicly available HCP data (see supplementary information in [61]). A total of 17 SI subregions were derived per cortical surface that included: Area 1: 1. Face, 2. Upper Extremity, 3. Trunk, 4. Lower Extremity; Area 2: 1. Upper extremity, 2. Trunk, 3. Lower Extremity; Areas 3a and 3b: 1. Face, 2. Ocular, 3. Upper Extremity, 4. Trunk, 5. Lower Extremity. Thus, the HCP_MMP1.0_SIsubregions parcellation contains 193 cortical parcels per hemisphere for a total of 386 parcels, compared to 180 cortical parcels per hemisphere in the original HCP_MMP1.0 parcellation (four original SI parcels removed and 17 SI subregions added per hemisphere).

Since this cortical parcellation excludes subcortical structures, we added the latter by using a recently published subcortical atlas that was developed based on functional connectivity gradients from > 1,000 HCP Young Adult (HCP-YA) participants data [63], named Melbourne Subcortex Atlas (MSA). We chose the scale IV of this subcortical atlas, which includes subdivisions of well-known anatomical nuclei, namely, the amygdala, hippocampus, thalamus, globus pallidum (GP), nucleus accumbens (NAc), caudate and the putamen, as well as an anteroposterior thalamic partition, totaling 27 subcortical regions per side. Finally, the periaqueductal gray (PAG) was added to the subcortical atlas based on a probabilistic mapping of this structure [64]. This way, this cortical and subcortical parcellated scheme (HCP_MMP1.0_SIsubregions+MSA-PAG) includes 386 cortical parcels plus 55 subcortical regions, totaling 441 regions of interest (ROIs).

The GLM design matrix for level 1 analysis (individual stimfMRI scan) included one explanatory variable (EV) of interest for the dentoalveolar pressure stimulation, modeled with a square waveform for the stimulus ON/OFF timing; and a nuisance EV for the temporal derivative of the stimulation waveform. A high-pass filter of 60 s was used to remove low frequency artifacts and no additional smoothing was done, so that the total smoothing level for stimfMRI time series was 2 mm full width at half maximum (FWHM). Level 1 analysis implemented temporal autocorrelation correction and model estimation by convolving the EVs with a double-gamma hemodynamic response function. Level 2 analysis (within-participant) combined both stimfMRI runs for each participant using a fixed-effects model. Additional steps needed prior to group analysis included computing cortical surface average area for each participant in order to perform permutation analysis for cortical surface statistical inference, and making an average dataset including all participants' data so that stimfMRI data could be viewed on the group average cortical surface and brain volume files within wb_view (part of Connectome Workbench).

Group analysis (level 3) was performed using non-parametric permutation methods provided by Permutation Analysis of Linear Models (PALM) (version alpha119) [65], an ancillary FSL tool. Input files were contrasts of parameter estimates (COPE) for dentoalveolar pressure

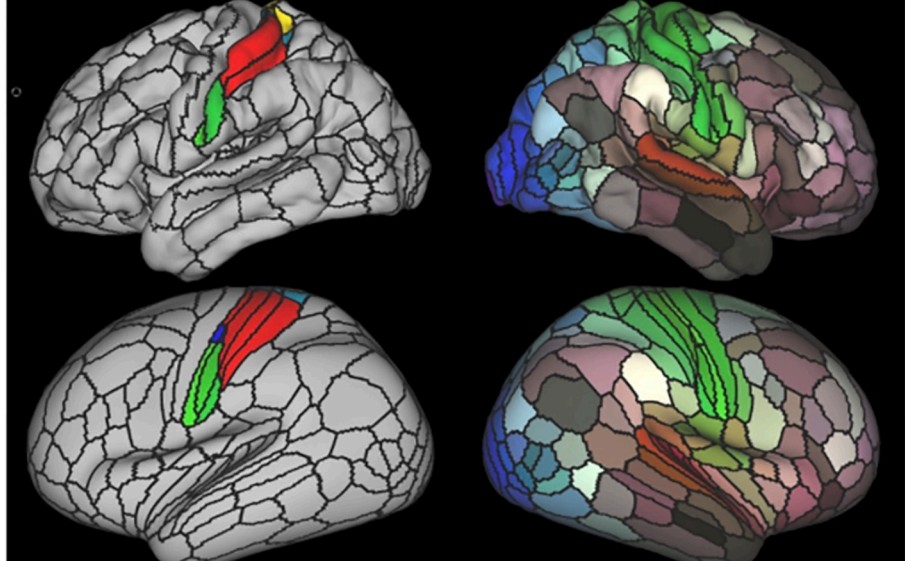

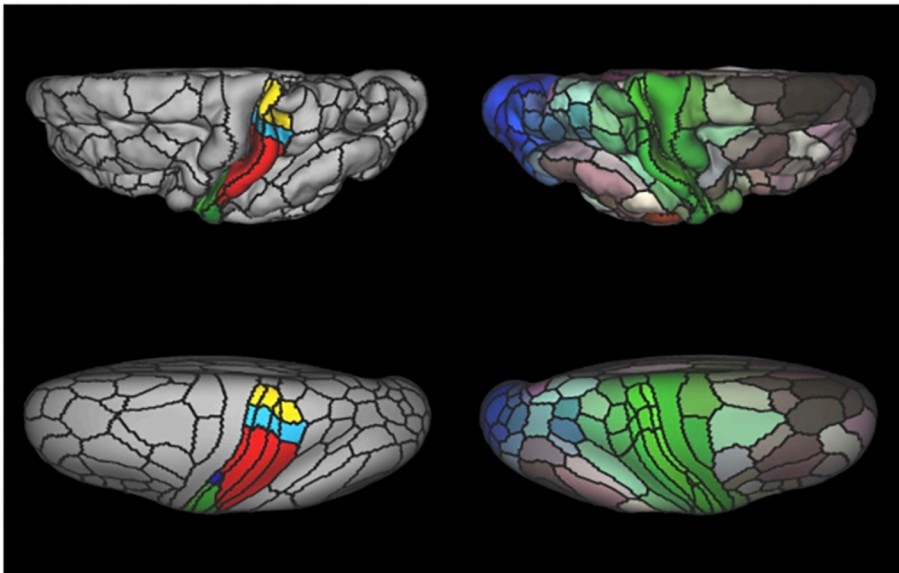

**Fig 2. HCP_MMP1.0_modified-S1subregions parcellation.** Pial surface cortical views are shown in the first row of each panel. Very inflated cortical views are shown in the second row of each panel. Black lines delineate individual parcel borders. Brain images on the left show the left hemisphere from lateral (a) and top-down (b) views; images on the right show the same views of the right hemisphere. In left hemisphere views, SI is divided into somatotopic subregions by color (green: Face, dark blue: Eye, red: Upper limb, light blue: Torso, yellow: Lower limb). In right hemisphere views, colors illustrate the extent to which cortical areas are connected in the resting state to auditory (red), somatosensory (green), visual (blue), task-positive (paler shades), and task-negative (darker shades) systems.

stimulation generated by the level 2 analysis described above. All statistical tests were done by performing 5,000 permutations for each test. One-way independent ANOVA followed by pairwise comparison testing were performed to assess between-group differences for mean group activations following dentoalveolar pressure stimulation. Within-group mean brain activations/deactivations were determined using a one-sample t-test GLM model.

For dense stimfMRI data, PALM was run independently on the brain cortical and subcortical data components of the COPE CIFTI files since we planned to use a spatial statistic, namely

threshold-free cluster enhancement (TFCE). The cortical processing also included the pre-calculated individual cortical surface average area files to better represent the sizes of clusters and of the support regions of TFCE. Permutation test results produced family-wise error rate (FWER)-corrected p-values for the TFCE statistic to correct for multiple testing across cortical surface vertices and subcortical voxels for each component, respectively. These p-values were output as -$\log_{10}$(p-value) as it offers better contrast for its visualization. Output files from processing each component were subsequently merged back into a single CIFTI file for visual assessment. To correct for multiple comparisons across the two sets of results (cortical, subcortical) merged, we applied the Sidak correction method (-$\log_{10}$ (1-(1-p-value)^(1/N))) so that only results with values $\geq$ 1.597 (corresponding to p-value $\leq$ 0.025) were considered significant when visualized in wb_view. Assignment of significantly activated clusters to brain regions was done by using wb_view to visually assess their locations relative to the HCP_MMP1.0_SIsubregions+MSA-PAG, with the label file serving as an anatomical guide for the ROIs included in it.

For parcellated stimfMRI data, PALM was run on both cortical and subcortical components simultaneously using the input COPE CIFTI files containing the 441 ROIs from the HCP_MMP1.0_SIsubregions+MSA-PAG. Since parcellated analyses do not use spatial statistics such as the TFCE, there is no need to separate the cortical and subcortical components. Permutation test results were FWER-corrected for multiple comparisons across all ROIs, and p-values were also output as -$\log_{10}$(p-value). Since no separation of the cortical and subcortical components is done for parcellated analysis, no Sidak correction is needed and activations with values of -$\log_{10}$(0.05) = 1.301 and greater were considered significant.

All comparisons (between-groups, within-group) used two-tailed permutation tests. Cohen's d effect sizes in each ROI were calculated for group mean results using PALM. Spatial maps were then thresholded in wb_view to show effect sizes greater than 0.8 or less than -0.8, which are considered large by Cohen's interpretation [66]. This approach allowed for visualization of both significant brain activations and deactivations for those comparisons in response to the noxious dentoalveolar stimulus, as shown by positive and negative effect sizes, respectively.

Finally, exploratory group analyses were conducted by adding potential covariates of interest, including the scores for PHQ-9 (depression), GAD-7 (anxiety), OBC (oral behaviors), PHQ-15 (somatic symptoms), PSS (perceived stress) and PSQI (sleep quality) to determine if these covariates would influence mean group brain activations for the chronic TMD low- and high-impact pain groups.

## Results

Fifty-two participants were enrolled in the main study protocol; however, two were excluded from the present analysis as stimfMRI scans were not acquired because of technical problems during the neuroimaging session. Thus, fifty age-matched females completed the study and had all neuroimaging data acquired, with the following group distribution and GCPS grades: controls (n = 17; Grade 0 = 15, Grade I = 2); low-impact TMD (n = 17; Grade I = 12, Grade IIa = 5); and high-impact TMD (n = 16; Grade IIb = 4, Grade III = 9, Grade IV = 3). The two controls with non-zero scores for the CPI reported experiencing sinus pain in the six months (time frame used in the GCPS for CPI-related items) prior to study participation but denied any pain in the TMJs, masticatory muscles and associated headaches–thus still fulfilling inclusion criteria for pain-free controls.

Dentoalveolar noxious stimulation sites were matched across groups, though more participants were stimulated on the left side (*Maxillary Right*: 4 controls, 3 low-impact TMD, 2 high-

impact TMD; *Mandibular Right*: 3 controls, 4 low-impact TMD, 3 high-impact TMD; *Maxillary Left*: 4 controls, 5 low-impact TMD, 5 high-impact TMD; *Mandibular Left*: 6 controls, 5 low-impact TMD, 6 high-impact TMD).

The time between visits two and three was of most interest for the purposes of our study since: 1. The dentoalveolar pressure stimulus was determined in visit 2; and 2. The neuroimaging session where the dentoalveolar pressure stimulation took place during the third visit. This interval (in mean number of days ± SD) was similar across the three groups as follows: controls (3.9 ± 1.8), low-impact TMD (4.3 ± 2.1), and high-impact TMD (3.9 ± 1.5).

## Participant characteristics

Anthropometric, psychosocial, and clinical pain characteristics are outlined in Table 1. Age, body mass index (BMI) category, and handedness were similar across groups. Sociodemographic characteristics including ethnicity, marital status, education, and household income were similar across groups. Participant race was distributed similarly across groups with the exception being there were four Asian participants in the control group and none in either TMD group. High-impact TMD pain participants scored significantly higher than controls in all psychosocial questionnaires, indicating worse outcomes for the included measures. Participants in the low-impact TMD pain group also scored higher than controls in most questionnaires (Table 1). Only JFLS-20 scores differed significantly between TMD groups, with the high-impact TMD pain group reporting greater impairment in jaw function (p = 0.048). TMD pain groups did not differ significantly in jaw pain duration, number of painful body sites, or comorbidity index.

DC/TMD clinical diagnoses for TMD participants are displayed in Fig 3. All but one TMD case had additional diagnoses other than myalgia.

## Pain sensitivity confounders

Pain sensitivity confounders such as time of day of the visit, menstrual cycle phase, caffeine intake in the previous 24 hours, and mean MQS score were similar for all groups prior to the neuroimaging session (Table 1). Measures of current jaw pain intensity and GBS pain intensity and unpleasantness ratings were similar for TMD groups on the day of the neuroimaging session (Table 1).

## Dentoalveolar pressure pain ratings for stimfMRI

Stimulus pain ratings were recorded following each stimfMRI run as described in the methods section using a 0–10 NRS, and both ratings were averaged so that each participant had a single rating for pain intensity secondary to the overall dentoalveolar pressure stimulation during stimfMRI. Although average pain ratings increased according to the TMD pain impact status (controls: 3.76 ± 1.88 [mean ± SD]; low-impact TMD: 4.65 ± 1.95; high-impact TMD: 5.34 ± 2.03), no significant between-group differences were found using Games-Howell post-hoc tests (controls vs. low-impact TMD, p = 0.383; controls vs. high-impact TMD, p = 0.069; low-impact TMD vs. high-impact TMD, p = 0.580).

## Dense stimfMRI results

Dense stimfMRI data analysis revealed no significant between-group brain activation differences following dentoalveolar pressure pain stimuli. Within-group mean activation analysis showed no significant mean brain activations for controls. The low-impact TMD group exhibited significant activations within the right hemisphere only, while bilateral, though right

**Table 1. Anthropometric, psychosocial, and pain characteristics.**

| Participant Characteristics | Controls (n = 17) | Low-impact TMD (n = 17) | High-impact TMD (n = 16) | P-value | Post-hoc pairwise comparisons[c] (P-value) | | |
|---|---|---|---|---|---|---|---|
| | | | | | Controls vs. low-impact TMD | Controls vs. high-impact TMD | Low- vs. high-impact TMD |
| Anthropometric | | | | | | | |
| Age (y) | 34.5 ± 13.7 | 37 ± 15.8 | 35.7 ± 12.0 | | 0.867 | 0.955 | 0.961 |
| BMI (kg/m$^2$) | 25.7 ± 6.5 | 26.9 ± 6.3 | 27.4 ± 7.9 | 0.917[a] | | | |
| Handedness (self-report) | Left-handed = 0 | Left-handed = 2 | Left-handed = 2 | 0.448[a] | | | |
| | Right-handed = 17 | Right-handed = 15 | Right-handed = 14 | | | | |
| Psychosocial | | | | | | | |
| GCPS Grade | Grade 0 = 15 Grade I = 2 | Grade I = 12 Grade IIa = 5 | Grade IIb = 4 Grade III = 9 Grade IV = 3 | | | | |
| JFLS-20 (0–10) | 0.0 ± 0.0 | 1.9 ± 1.1 | 3.0 ± 1.5 | | < .001 | < .001 | 0.048 |
| PHQ-9 (0–27) | 1.8 ± 1.5 | 6.5 ± 5.2 | 6.3 ± 5.3 | | 0.005 | 0.012 | 0.987 |
| GAD-7 (0–21) | 3.1 ± 3.8 | 6.5 ± 5.6 | 7.8 ± 6.5 | | 0.057 | 0.049 | 0.810 |
| PHQ-15 (0–30) | 3.7 ± 2.7 | 8.9 ± 4.3 | 10.4 ± 6.8 | | < .001 | 0.004 | 0.739 |
| OBC (0–84) | 18.9 ± 7.0 | 34.5 ± 10.4 | 34.7 ± 16.3 | | < .001 | 0.005 | 0.999 |
| PSS (0–40) | 10.7 ± 6.4 | 15.9 ± 7.1 | 17.6 ± 6.6 | | 0.077 | 0.011 | 0.746 |
| PSQI (0–21) | 4.0 ± 2.4 | 7.8 ± 4.5 | 8.9 ± 3.4 | | 0.012 | < .001 | 0.727 |
| Clinical Pain Measures (Visit 1) | | | | | | | |
| GCPS Characteristic Pain Intensity (0–100) | 3.3 ± 10.8 | 46.3 ± 15.4 | 66.7 ± 8.94 | | < .001 | < .001 | < .001 |
| Jaw pain duration (months) | — | 173 ± 141 | 128 ± 107 | 0.441[b] | | | |
| Total number of painful body sites (0–45) | 0.5 ± 1.2 | 13.4 ± 7.9 | 14.2 ± 9.0 | | < .001 | < .001 | 0.963 |
| Comorbidity Index (0–18) | 0.4 ± 1.0 | 2.5 ± 2.0 | 2.6 ± 1.9 | | 0.002 | 0.001 | 0.999 |
| Pre-MRI Pain Measures (Visit 3) | | | | | | | |
| GBS pain intensity (0–20) | — | 6.4 ± 4.2 | 8.6 ± 4.0 | 0.095[d] | | | |
| GBS pain unpleasantness (0–20) | — | 5.8 ± 3.5 | 7.5 ± 3.4 | 0.170[d] | | | |
| Current jaw pain intensity (0–100) | — | 19.1 ± 19.3 | 22.9 ± 16.4 | 0.292[b] | | | |
| Pain sensitivity confounders (Visit 3) | | | | | | | |
| Time of day of study visit | Morning = 9 | Morning = 7 | Morning = 10 | 0.527[a] | | | |
| | Afternoon = 8 | Afternoon = 10 | Afternoon = 6 | | | | |
| Menstrual cycle phase | No menses = 4 | No menses = 7 | No menses = 7 | 0.841[a] | | | |
| | Menstrual = 1 | Menstrual = 2 | Menstrual = 1 | | | | |
| | Follicular = 1 | Follicular = 2 | Follicular = 1 | | | | |
| | Periovulatory = 2 | Periovulatory = 1 | Periovulatory = 1 | | | | |
| | Luteal = 7 | Luteal = 2 | Luteal = 5 | | | | |
| | Premenstrual = 2 | Premenstrual = 2 | Premenstrual = 1 | | | | |
| | Missing = 0 | Missing = 1 | Missing = 0 | | | | |
| Caffeine intake last 24 h | None = 6 | None = 6 | None = 9 | 0.295[a] | | | |
| | Low = 5 | Low = 4 | Low = 6 | | | | |
| | Moderate = 3 | Moderate = 2 | Moderate = 1 | | | | |
| | High = 3 | High = 5 | High = 0 | | | | |

*(Continued)*

**Table 1.** (Continued)

| Participant Characteristics | Controls (n = 17) | Low-impact TMD (n = 17) | High-impact TMD (n = 16) | P-value | Post-hoc pairwise comparisons[c] (P-value) | | |
|---|---|---|---|---|---|---|---|
| | | | | | Controls vs. low-impact TMD | Controls vs. high-impact TMD | Low- vs. high-impact TMD |
| MQS score | 0.71 ± 1.63 | 3.98 ± 5.14 | 2.69 ± 5.38 | | 0.055 | 0.355 | 0.766 |

BMI, body mass index; GCPS, Graded Chronic Pain Scale; JFLS-20, Jaw Function Limitation Scale 20-items; PHQ-9, Patient Health Questionnaire-9; GAD-7, Generalized Anxiety Disorder-7; PHQ-15, Patient Health Questionnaire-15; OBC, Oral Behavior Checklist; PSS, Perceived Stress Scale; PSQI, Pittsburgh Sleep Quality Index; Comorbidity index was based on the self-reported presence of 18 comorbid conditions; GBS, Gracely Box Scale; MQS, Medication Quantification Scale.

Mean ± SD.

[a]Fisher's exact test.

[b]Mann-Whitney U test.

[c]Games Howell test.

[d]Independent samples *t* test.

hemisphere-dominant, cortical activations were seen in the high-impact group (Fig 4) Two significant activation clusters were found for each TMD group, but the high-impact TMD group showed a spatially broader overall activation (17,083 vertices) than the low-impact group (4,512 vertices). In the high-impact TMD group, the right hemisphere activation cluster involved 16,938 vertices overlapping numerous cortical regions, including the somatosensory and motor, premotor, posterior opercular, early auditory, insular and frontal opercular, temporo-parieto-occipital junction, inferior parietal, inferior frontal, and dorsolateral prefrontal cortices. The small left hemisphere activation cluster in the high-impact TMD group (145 vertices) overlapped with the posterior opercular and inferior parietal cortices. In the low-impact group, the larger activation cluster (4,398 vertices) involved the right somatosensory and motor, posterior opercular, early auditory, and inferior parietal cortices. The smaller cluster (114 vertices) overlapped the right premotor cortex.

Significant subcortical activations were also found for both TMD groups, although with relatively small cluster size (largest cluster for low-impact TMD: 153 voxels; high-impact TMD:

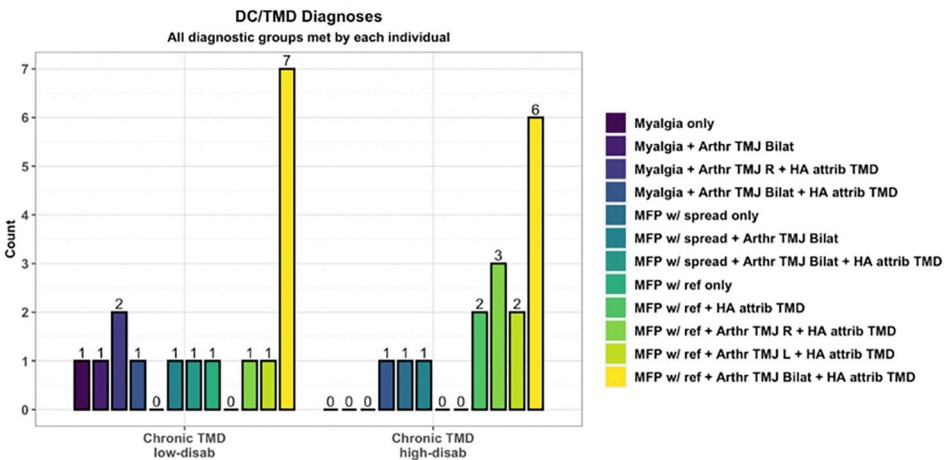

**Fig 3. DC/TMD diagnoses for chronic TMD pain cases.** Arthr = arthralgia; MFP w/ ref = myofascial pain with referral; HA attrib TMD = headache attributed to TMD.

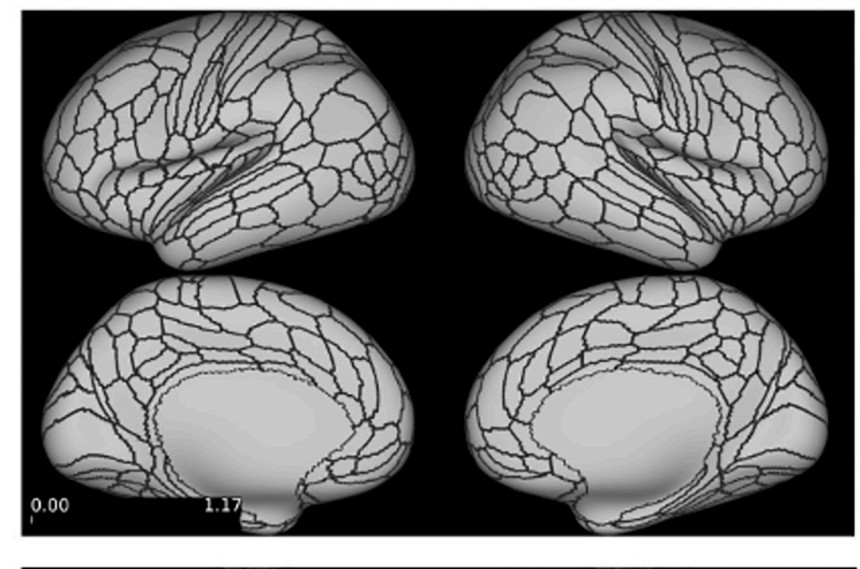

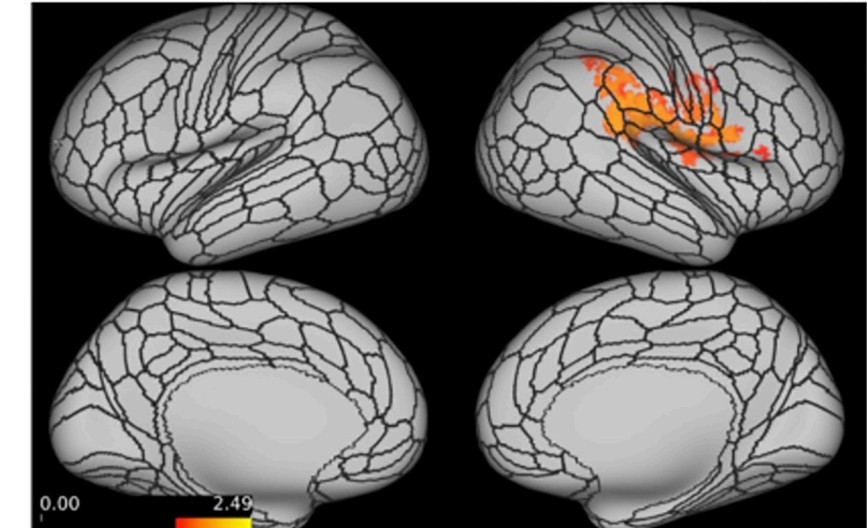

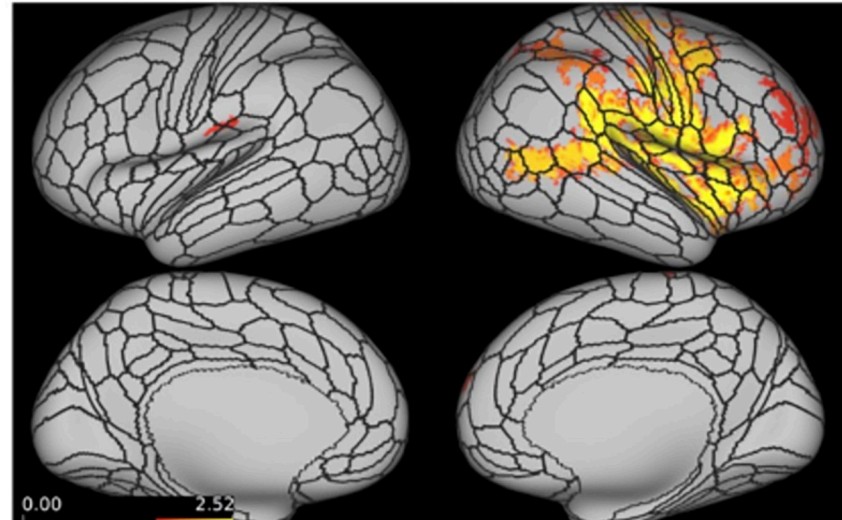

**Fig 4. Dense stimfMRI cortical activations.** Dense stimfMRI cortical activations for the controls (a), low-impact (b) and high-impact (c) TMD pain groups (no activations were found in control group). Red-yellow colors indicate the range of significant cortical activations in (-$\log_{10}$ (p-value)) format following Sidak correction. The minimum threshold for visualization was set at 1.597 (red color), corresponding to p = 0.025. The color bars indicate the range of p-values, with higher and lower p-values depicted in a gradient of red to yellow, respectively. Vertices with log transformed p-values $\geq$ 1.597 following a two-tailed t-test appear activated.

25 voxels). Twenty clusters were found for the low-impact TMD group, ten of them with $\geq$ 5 voxels, and all were located within the cerebellum. In the high-impact TMD group, seventeen clusters were found across subcortical structures, including activations in the putamen, thalamus, cerebellum, and caudate. There were seven clusters with $\geq$ 5 activated voxels in the high-impact TMD group (Table 2).

## Parcellated stimfMRI Results

Similar to dense stimfMRI results, no significant between-group differences were found in the parcellated stimfMRI analysis. Within-group mean activations are displayed in Fig 5, and Table 3 details the cortical parcels activated for each group and the respective brain regions to which those parcels belong. The HCP_MMP1.0_SIsubregions+MSA-PAG parcellation also assessed for subcortical ROI activations, with the only significant activation occurring in the right nucleus accumbens, core (NAc-core-rh: p = 0.010, Cohen's d = 1.13) of the high-impact TMD group.

The lone significant activation in the control group appeared in the right early auditory cortex (retro-insular cortex (RI) parcel: p = 0.008, d = 1.48) (Fig 5A). In the low-impact TMD group, activations within the posterior opercular cortex were found in right PFcm (p = 0.023,

**Table 2. Dense stimfMRI subcortical activations.**

| Group | Significantly Activated Clusters | Significant Activations (Voxels) | Max. MNI Coordinates (x, y, z) | Correlated Brain Region (Hemisphere) |
|---|---|---|---|---|
| Controls | 0 | - | | - |
| Low-impact TMD | 20 | 153 | (-24, -66, -48) | Cerebellum (L) |
| | | 17 | (-24, -60, -20) | Cerebellum (L) |
| | | 14 | (-32, -66, -42) | Cerebellum (L) |
| | | 12 | (14, -68, -46) | Cerebellum (R) |
| | | 11 | (-16, -80, -22) | Cerebellum (L) |
| | | 10 | (-26, -62, -28) | Cerebellum (L) |
| | | 9 | (28, -64, -54) | Cerebellum (R) |
| | | 9 | (-22, -74, -24) | Cerebellum (L) |
| | | 7 | (-18, -60, -18) | Cerebellum (L) |
| | | 6 | (-38, -62, -42) | Cerebellum (L) |
| High-impact TMD | 17 | 25 | (22, 8, -10) | PUT-VA (R) |
| | | 14 | (10, -20, 8) | THA-DAm (R) |
| | | 13 | (-8, -18, 8) | THA-DAm (L) |
| | | 9 | (-18, -66, -48) | Cerebellum (L) |
| | | 8 | (-14, -20, 8) | THA-VPl (L) |
| | | 6 | (16, -2, 18) | CAU-tail (R) |
| | | 5 | (-36, -58, -54) | Cerebellum (L) |

Dense sfMRI subcortical brain activations for all groups. For the TMD groups, only activation clusters including $\geq$ 5 voxels are included. All activations with a minimum Sidak corrected (-$\log_{10}$ (p-value)) value of 1.597 were deemed significant. PUT-VA = ventroanterior putamen; THA-DAm = medial dorsoanterior thalamus; THA-VPl = lateral ventroposterior thalamus; CAU-tail = caudate tail.

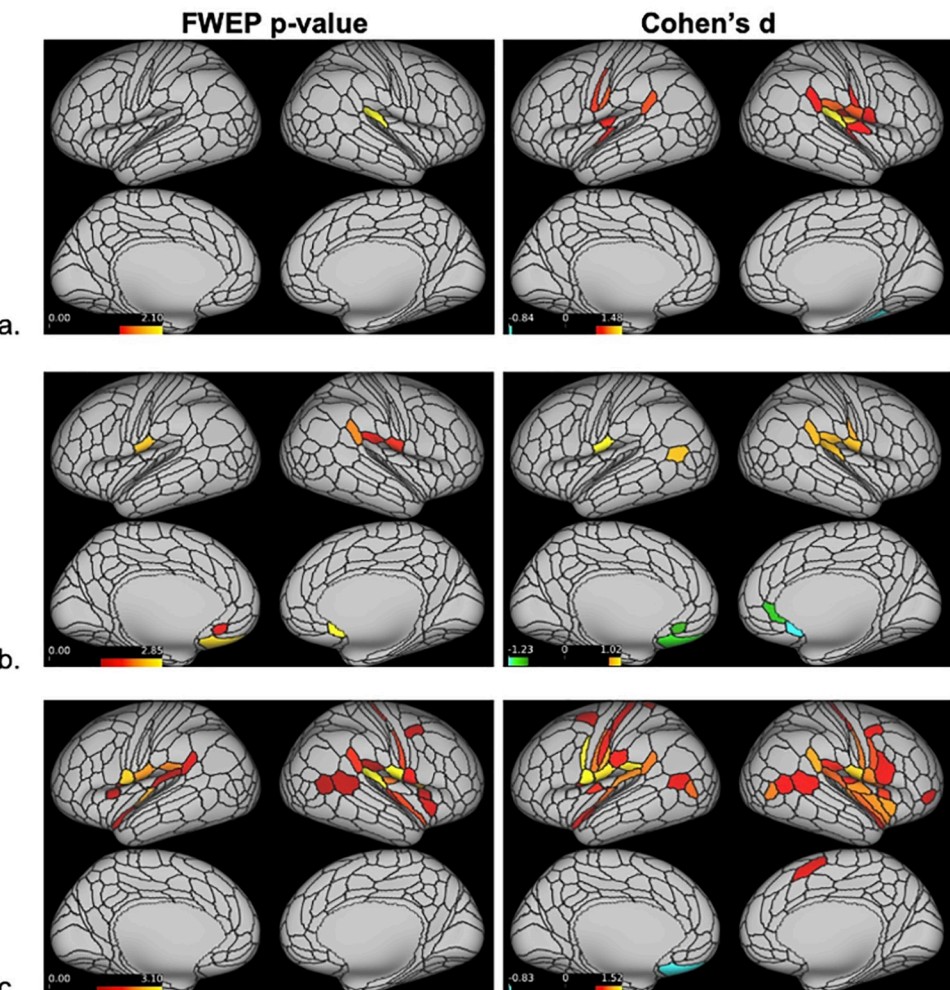

**Fig 5. Parcellated stimfMRI cortical activations.** Significantly activated/deactivated cortical parcels based on the HCP_MMP1.0_modified-SIsubregions parcellation (a. controls; b. low-impact TMD; c. high-impact TMD). On the left, parcels surviving a minimum threshold for visualization set at 1.301 (= -log₁₀(0.05)) following a two-tailed t-test appear activated. On the right, all parcels with large effect sizes, defined as Cohen's d ≥ 0.8 or ≤ -0.8, appear activated. Colors indicate the range of significant cortical activations/deactivations. Red-yellow colors represent significant activations, with yellow marking the "most significant/largest effect size" activations. Green-blue colors represent significant deactivations, with blue marking the "most significant/largest effect size" deactivations.

d = 0.97) and bilateral OP4 parcels (OP4-Left: p = 0.004, d = 1.10; OP4-right: p = 0.015, d = 1.00). The peri-sylvian-language (PSL) area, located within the temporo-parieto-occipital junction, was activated in the right hemisphere only (PSL: p = 0.008, d = 1.04). Only the low-impact TMD group presented significant deactivations, which appeared bilaterally within the anterior cingulate and medial prefrontal (s32-left: p = 0.028, d = -0.95; 25-right: p = 0.001, d = -1.16) and orbital and frontal cortices (OFC-left: p = 0.002, d = -1.13).

The high-impact TMD group exhibited the largest number of activated parcels distributed in six brain regions (Table 3, Fig 5C). Somatosensory and motor cortex activations appeared exclusively in the right hemisphere, involving SI somatotopic subregions representing the face and upper extremity (3aFace: p = 0.009, d = 1.11; 3aUpperExtremity: p = 0.011, d = 1.09). RI, the same early auditory cortex parcel activated in the control group, was also activated in the high-impact TMD group, albeit bilaterally (RI-left: p = 0.041, d = 0.98; RI-right: p = 0.001,

**Table 3. Parcellated stimfMRI cortical activations.**

| Hemisphere | [1]Brain Region | [1]Parcel | | Controls | Low-impact TMD | High-impact TMD |
|---|---|---|---|---|---|---|
| Left | #9 Posterior Opercular Cortex | OP4 | [2]Cohen's d | 0.77 | **1.10** | **1.16** |
| | | | [3]FWEP p-value | 0.126 | 0.004 | 0.004 |
| | | 43 | [2]Cohen's d | 0.64 | 0.71 | **1.26** |
| | | | [3]FWEP p-value | 0.429 | 0.344 | 0.002 |
| | | PFcm | [2]Cohen's d | 0.74 | 0.74 | **1.14** |
| | | | [3]FWEP p-value | 0.171 | 0.255 | 0.005 |
| | | FOP1 | [2]Cohen's d | 0.58 | 0.60 | **0.99** |
| | | | [3]FWEP p-value | 0.617 | 0.662 | 0.035 |
| | #10 Early Auditory Cortex | RI | [2]Cohen's d | 0.60 | 0.60 | **0.98** |
| | | | [3]FWEP p-value | 0.563 | 0.646 | 0.041 |
| | #12 Insular and Frontal Opercular Cortex | 52 | [2]Cohen's d | 0.44 | 0.36 | **1.19** |
| | | | [3]FWEP p-value | 0.973 | 0.999 | 0.003 |
| | | PI | [2]Cohen's d | 0.20 | -0.24 | **1.03** |
| | | | [3]FWEP p-value | 1.000 | 1.000 | 0.024 |
| | | FOP3 | [2]Cohen's d | 0.37 | 0.47 | **0.97** |
| | | | [3]FWEP p-value | 0.999 | 0.943 | 0.044 |
| | #15 Temporo-Parieto-Occipital Junction | PSL | [2]Cohen's d | 0.77 | 0.81 | **1.01** |
| | | | [3]FWEP p-value | 0.130 | 0.147 | 0.027 |
| | #19 Anterior Cingulate and Medial Prefrontal Cortex | s32 | [2]Cohen's d | -0.36 | **-0.95** | -0.20 |
| | | | [3]FWEP p-value | 0.999 | 0.028 | 1.000 |
| | #20 Orbital and Polar Frontal Cortex | OFC | [2]Cohen's d | -0.54 | **-1.13** | -0.78 |
| | | | [3]FWEP p-value | 0.762 | 0.002 | 0.244 |
| Right | #6 Somatosensory and Motor Cortex | 3aFace | [2]Cohen's d | 0.49 | 0.73 | **1.11** |
| | | | [3]FWEP p-value | 0.890 | 0.292 | 0.009 |
| | | 3aUpperExtremity | [2]Cohen's d | 0.23 | 0.30 | **1.09** |
| | | | [3]FWEP p-value | 1.000 | 1.000 | 0.011 |
| | #8 Premotor Cortex | 55b | [2]Cohen's d | 0.35 | 0.29 | **0.99** |
| | | | [3]FWEP p-value | 0.999 | 1.000 | 0.037 |
| | #9 Posterior Opercular Cortex | OP4 | [2]Cohen's d | 0.77 | **1.00** | **1.33** |
| | | | [3]FWEP p-value | 0.135 | 0.015 | 0.001 |
| | | PFcm | [2]Cohen's d | 0.80 | **0.97** | **0.96** |
| | | | [3]FWEP p-value | 0.099 | 0.023 | 0.048 |
| | | FOP1 | [2]Cohen's d | 0.46 | 0.73 | **1.19** |
| | | | [3]FWEP p-value | 0.999 | 0.295 | 0.003 |
| | | OP2-3 | [2]Cohen's d | 0.81 | 0.68 | **1.14** |
| | | | [3]FWEP p-value | 0.090 | 0.398 | 0.006 |
| | | 43 | [2]Cohen's d | 0.79 | 0.84 | **1.14** |
| | | | [3]FWEP p-value | 0.226 | 0.103 | 0.013 |
| | #10 Early Auditory Cortex | RI | [2]Cohen's d | **0.99** | 0.85 | **1.31** |
| | | | [3]FWEP p-value | 0.008 | 0.100 | 0.001 |
| | #12 Insular and Frontal Opercular Cortex | Pol1 | [2]Cohen's d | 0.39 | 0.72 | **1.11** |
| | | | [3]FWEP p-value | 0.995 | 0.316 | 0.009 |
| | | 52 | [2]Cohen's d | 0.50 | 0.67 | **1.09** |
| | | | [3]FWEP p-value | 0.876 | 0.446 | 0.012 |
| | | MI | [2]Cohen's d | 0.35 | 0.40 | **1.01** |
| | | | [3]FWEP p-value | 0.999 | 0.992 | 0.031 |

*(Continued)*

**Table 3.** (Continued)

| Hemisphere | [1]Brain Region | [1]Parcel | | Controls | Low-impact TMD | High-impact TMD |
|---|---|---|---|---|---|---|
| | | FOP3 | [2]Cohen's d | 0.52 | 0.81 | **0.97** |
| | | | [3]FWEP p-value | 0.816 | 0.147 | 0.046 |
| | #15 Temporo-Parieto-Occipital Junction | PSL | [2]Cohen's d | 0.62 | **1.04** | **1.10** |
| | | | [3]FWEP p-value | 0.479 | 0.008 | 0.011 |
| | | TPOJ1 | [2]Cohen's d | 0.02 | 0.49 | **0.97** |
| | | | [3]FWEP p-value | 1.000 | 0.913 | 0.047 |
| | #19 Anterior Cingulate and Medial Prefrontal Cortex | 25 | [2]Cohen's d | -0.61 | **-1.16** | -0.65 |
| | | | [3]FWEP p-value | 0.516 | 0.001 | 0.545 |

[1]Cortical regions and parcels defined by the HCP_MMP1.0_modified-SIsubregions parcellation.

[2]Significant large effect sizes (Cohen's d $\geq$ 0.8 or $\leq$ -0.8 with FWEP p-value $\leq$ .05) appear in bold.

[3]Cortical parcel log-transformed p-values (-log10(p-value)) were transformed back to p-values. Parcels with minimum value of 1.301 (= -log10(0.05)) in at least one group are reported.

*FWEP p-value $\leq$ .05 without large effect size.

d = 1.31). Similar to the low-impact TMD group, PSL was activated in high-impact TMD participants (PSL-left: p = 0.027, d = 1.01; PSL-right: p = 0.011, d = 1.23) along with posterior opercular cortex activations in bilateral OP4 (OP4-left: p = 0.004, d = 1.16; OP4-right: p = 0.001, d = 1.10) and right PFcm (p = 0.048, d = 0.96). Other activations in this region seen only in the high-impact TMD group included bilateral areas 43 (43-left: p = 0.002, d = 1.26; 43-right: p = 0.013, d = 1.14) and FOP1 (FOP1-left: p = 0.035, d = 0.99; FOP1-right: p = 0.003, d = 1.19), left PFcm (p = 0.005, d = 1.14), and right OP2-3 (p = 0.006, d = 1.14). Unlike the other groups, high-impact TMD participants showed numerous activations within the insular and frontal opercular cortex: area 52 bilaterally (52-left: p = 0.003, d = 1.19; 52-right: p = 0.012, d = 1.09), left parainsular cortex (PI: p = 0.024, d = 1.03), bilateral frontal opercular area 3 (FOP3-left: p = 0.044, d = 0.97; FOP3-right: p = 0.046, d = 0.97), right posterior insula 1 (PoI1: p = 0.009, d = 1.11), and the right middle insula (MI: p = 0.031, d = 1.11). Also unique to the high-impact TMD group were activations within the right premotor cortex (55b: p = 0.037, d = 0.99) and temporo-parieto-occipital junction (TPOJ1-right: p = 0.047, d = 0.97).

Because the majority of participants were stimulated in left-sided intraoral quadrants (66%), a secondary parcellated analysis was conducted using data from these participants only to assess brain activity related to unilateral stimuli (Table 4, Fig 6).

No significant cortical activations were present for the control (n = 10) or low-impact TMD groups (n = 10). The lone significant deactivation in this analysis occurred in area 25 of the low-impact TMD group (25-right: p = 0.004, d = -1.35). In the high-impact TMD group (n = 11), left hemisphere activations occurred in the insular and frontal opercular cortex (52: p = 0.047, d = 1.14) and right hemisphere activations were seem in the somatosensory and motor cortex (3aFace: p = 0.014, d = 1.29; 3aUpperExtremity: p = 0.022, d = 1.25), premotor cortex (55b: p = 0.024, d = 1.22), posterior opercular cortex (FOP1: p = 0.013, d = 1.30; OP4: p = 0.023, d = 1.24), and early auditory cortex (RI: p = 0.007, d = 1.40).

In addition to the Cohen's d values reported above, effect sizes for pairwise comparisons were calculated for between-group brain activations differences. A relatively wide range of effect sizes was detected across these comparisons (controls vs. high-impact TMD (Cohen's d range: -0.568 to 1.225; p $\geq$ 0.286); controls vs. low-impact TMD (-0.721 to 0.818; p $\geq$ 0.985); low-impact TMD vs. high-impact TMD (-0.552 to 1.268; p $\geq$ 0.219)). Despite certain large effect sizes, pairwise comparisons failed to reach significance thresholds for any parcels meaning that no significant between-group differences were found.

**Table 4. Parcellated stimfMRI cortical activations–left-sided stimulus only.**

| Hemisphere | [1]Brain Region | [1]Parcel | | Controls | Low-impact TMD | High-impact TMD |
|---|---|---|---|---|---|---|
| Left | #12 Insular and Frontal Opercular Cortex | 52 | [2]Cohen's d | 0.20 | 0.33 | **1.14** |
| | | | [3]FWEP p-value | 1.000 | 1.000 | 0.047 |
| Right | #6 Somatosensory and Motor Cortex | 3aFace | [2]Cohen's d | 0.26 | 0.58 | **1.29** |
| | | | [3]FWEP p-value | 1.000 | 0.875 | 0.014 |
| | | 3aUpperExtremity | [2]Cohen's d | 0.33 | 0.26 | **1.25** |
| | | | [3]FWEP p-value | 1.00 | 1.000 | 0.022 |
| | #8 Premotor Cortex | 55b | [2]Cohen's d | 0.49 | 0.42 | **1.22** |
| | | | [3]FWEP p-value | 0.993 | 0.996 | 0.024 |
| | #9 Posterior Opercular Cortex | FOP1 | [2]Cohen's d | 0.16 | 0.71 | **1.30** |
| | | | [3]FWEP p-value | 1.000 | 0.606 | 0.013 |
| | | OP4 | [2]Cohen's d | 0.48 | 0.84 | **1.24** |
| | | | [3]FWEP p-value | 0.997 | 0.295 | 0.023 |
| | #10 Early Auditory Cortex | RI | [2]Cohen's d | 0.79 | 0.77 | **1.40** |
| | | | [3]FWEP p-value | 0.492 | 0.459 | 0.007 |
| | #19 Anterior Cingulate and Medial Prefrontal Cortex | 25 | [2]Cohen's d | -0.90 | **-1.35** | -0.62 |
| | | | [3]FWEP p-value | 0.261 | 0.004 | 0.842 |

[1]Cortical regions and parcels defined by the HCP_MMP1.0_modified-SIsubregions parcellation

[2]Significant large effect sizes (Cohen's d $\geq$ 0.8 or $\leq$ -0.8 with FWEP p-value $\leq$ .05) appear in bold.

[3]Cortical parcel log-transformed p-values (-log10(p-value)) were transformed back to p-values. Parcels with minimum value of 1.301 (= -log10(0.05)) in at least one group are reported.

Exploratory analyses using potential covariates of interest did not show statistically significant results for either chronic TMD pain group.

## Discussion

The present study used stimfMRI to assess brain responses secondary to noxious dentoalveolar stimuli in chronic TMD pain patients and matched controls. Counter to our expectations, there were no significant between-group differences in either the dense or parcellated stimfMRI analyses. At the within-group level for the dense analysis, both chronic TMD pain groups displayed significant cortical and subcortical mean brain activations, while controls did not. Similarly for the parcellated analysis, chronic TMD pain groups showed a substantially greater number of significantly activated cortical parcels than controls, with the greatest number found in the high-impact TMD group.

The lack of between-group differences for brain activation following dentoalveolar stimulation could be explained by several factors. Mean effect sizes for within-group activations were similar across groups in the parcellated analysis, indicating that meaningful activation differences were minimal (Table 3). Additionally, the body area stimulated by the noxious stimuli was relatively small (2 mm$^2$ contacting surface, 5–7 mm excursion over dentoalveolar tissues). It is possible that the receptive field of the trigeminal primary afferent neurons innervating this area is relatively large, in which case the dentoalveolar stimulation would evoke a limited neuronal response in their terminations within the CNS despite being perceived as painful. Our study's sample size (n < 20 for all groups) was also relatively small, likely making it more difficult to detect significant between-group differences.

Within-group mean brain activations presented a gradient activation pattern that paralleled the pain impact status; that is, the least number of activations were seen for pain-free controls

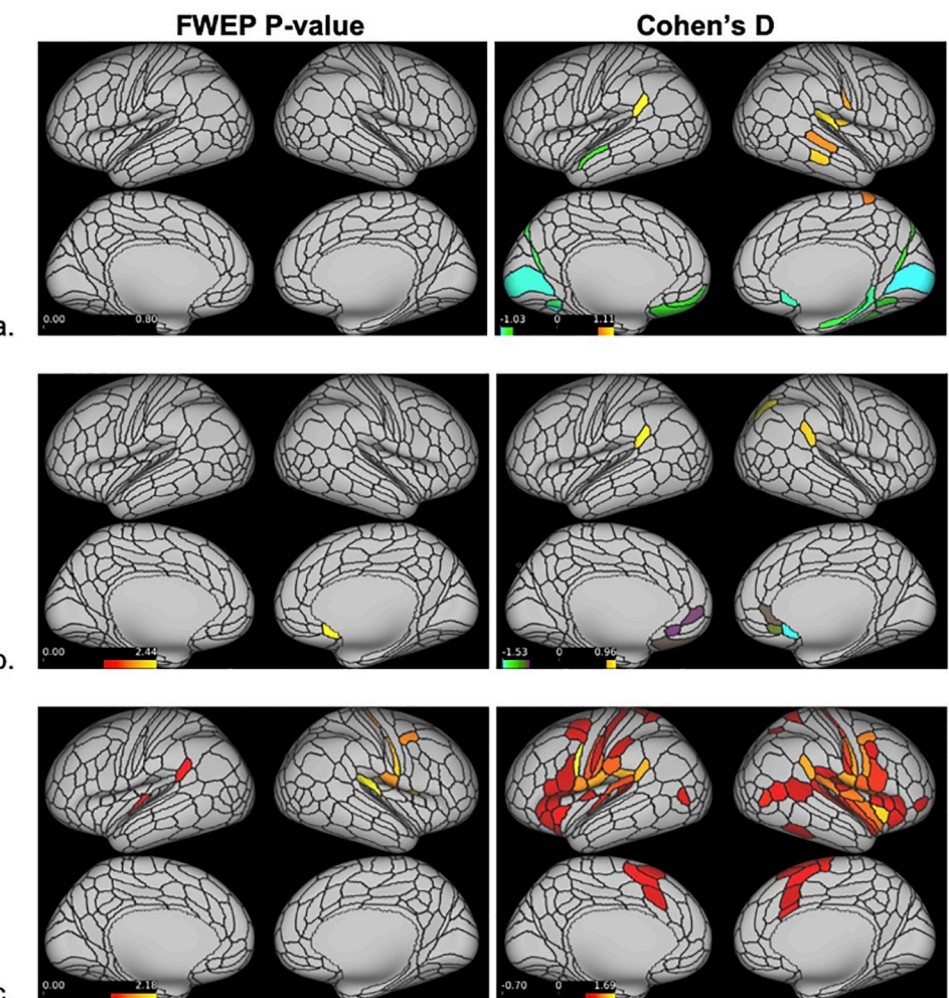

**Fig 6. Parcellated stimfMRI cortical activations–left-sided stimulus only.** Significantly activated/deactivated cortical parcels in response to left-sided only dentoalveolar pressure stimulus, based on the HCP_MMP1.0_modified-SIsubregions parcellation (a. controls; low-impact TMD; b. high-impact TMD). On the left, parcels surviving a minimum threshold for visualization set at 1.301 (= -log₁₀(0.05)) following a two-tailed t-test appear activated. On the right, all parcels with large effect sizes, defined as Cohen's d ≥ 0.8 or ≤ -0.8, appear activated. Colors indicate the range of significant cortical activations/deactivations. Red-yellow colors represent significant activations, with yellow marking the "most significant/largest effect size" activations. Green-blue colors represent significant deactivations, with blue marking the "most significant/largest effect size" deactivations.

and the greatest for high-impact TMD cases. The mean BOLD response for the high-impact TMD group was generally the greatest, which seems to partially explain the results, as the magnitude of the mean BOLD response is reflected in the effect sizes reported (Cohen's d > 0.8). However, many parcels in our study with large effect sizes were not statistically significant based on their p-values. Instead, we found that significantly activated parcels were associated with effect sizes of d ≥ 0.95 (or d ≤ -0.95) (Table 3). It is known that with small samples, effect sizes are typically inflated and can be inconsistent, so we interpret these findings to suggest that traditional Cohen's d thresholds (0.2 = small; 0.5 = medium; 0.8 = large) may not apply to our sample set.

The subjective pain ratings following the stimfMRI scans also showed this gradient pattern, with a trend towards significance for the high-impact TMD group reporting greater mean

pain ratings compared to controls (p = 0.069). As the three groups were matched for age, sex, intraoral quadrant stimulated, and subjective stimulus intensity, this finding suggests a potential involvement of increased pain facilitation and/or deficient pain modulation in females with chronic TMD pain, particularly as pain-impact worsens.

When assessing brain activations using stimfMRI, multimodal parcellation of the cerebral cortex and subcortical structures increases statistical sensitivity and power in neuroimaging analysis by increasing signal-to-noise ratio (SNR) and reducing the number of statistical comparisons required [52]. Thus, significant ROIs determined from parcellated stimfMRI analyses are relatively easier to interpret than activation clusters from dense stimfMRI analyses. Consequently, here we place a greater focus on the parcellated stimfMRI analysis results when discussing the cortical brain responses to dentoalveolar pressure pain stimuli as found in the present study.

### stimfMRI subcortical analysis

TMD groups differed in the volume and size of subcortical clusters as well as the specific regions activated. The low-impact TMD group exhibited more activations than the high-impact group, and the cluster sizes were generally larger. All activations in the low-impact TMD group occurred within the cerebellum, while numerous regions were activated in the high-impact group, including the cerebellum, putamen, caudate, and thalamus.

The cerebellum, putamen, and caudate are involved with the motor response to pain [32,67]. The cerebellum is commonly activated in both healthy individuals and chronic pain patients following noxious stimuli [68,69]. Findings related to the role of the putamen in chronic TMD pain are heterogenous [20,35], though alterations in structure and function have been reported [12,26]. One study showed reduced resting-state functional connectivity between the caudate and putamen in patients with myofascial TMD pain [26]. The same study also reported reduced functional connectivity between the putamen and the precentral gyrus in these patients, suggesting dysfunction in networks related to motor control. In our study, the presence of TMD group mean activations in subcortical regions associated with sensorimotor activity may indicate that alterations in these regions are involved in dentoalveolar pain processing in chronic TMD pain. The exact mechanisms by which pain and motor networks interact in TMD pain patients requires further investigation in future studies.

The thalamus is a critical brain region in trigeminal nociceptive processing [70]. In TMD patients, studies have revealed increased thalamic activations in response to non-noxious tactile stimulation of the index finger [25] and increased gray matter volume in left ventral posterior and right ventral lateral thalamus [20], reflecting neuroplasticity along the trigeminothalamocortical pathways. A meta-analysis of fMRI studies applying noxious stimuli to trigeminal structures in patients with several chronic orofacial pain conditions, including TMD, reported consistently increased activation in medial and posterior thalamus in patients compared to controls [12], offering evidence of maladaptive central pain modulation in these patients. While no between-group differences were detected in our study, significant thalamic mean activations were only seen in the high-impact TMD group within bilateral medial dorsoanterior nuclei and, to a lesser degree, the left lateral ventroposterior nucleus. Ascending secondary trigeminal afferents, including those from nociceptors, are known to synapse within the ventral posteromedial (VPM) nucleus of the thalamus [71], but other thalamic nuclei play a role in trigeminal pain processing. For example, an fMRI study applying noxious electrical pulp stimuli to upper and lower canines of healthy volunteers found increased activations in bilateral medial dorsal nuclei within the thalamus [72]. Our findings suggest that thalamic activity in response to dentoalveolar noxious stimuli may be differentially modulated relative to TMD pain-impact status.

The only subcortical activation in the parcellated analysis was the right nucleus accumbens, core (NAc-core-rh) of the high-impact TMD group. Historically, the NAc is known for its role in pleasure and reward, but studies have also highlighted the region's involvement in the transition from acute to chronic pain [73,74].

## stimfMRI cortical analysis

The lack of a meaningful number of mean brain activations for the control group was unexpected, as brain regions fundamental for acute pain processing in healthy individuals have been previously described [75]. Although the pain-free participants reported moderate subjective pain following the stimfMRI runs, the evoked mean BOLD responses were not strong enough to survive statistical thresholding corrected for multiple comparisons. This was not an impediment for either of the TMD groups, where several brain regions presented p-values below the FWER-correct threshold, despite comparable sample sizes across all three groups.

Among TMD pain groups, both dense and parcellated analyses showed a predominance of right hemisphere cortical activations. The dentoalveolar stimulation was applied to all four intraoral quadrants in a matched design across groups, but there was an uneven distribution between left- and right-sided stimulation across the three groups (left—31, right—19) which may have contributed to this finding. However, it has been reported that unilateral noxious trigeminal input activates bilateral ventral posterior thalamic nuclei and bilateral SI and SII face subregions in healthy subjects [76]. The study reported that innocuous unilateral lower lip stimuli only activated the contralateral thalamus. As previously mentioned, we observed bilateral thalamic activations in the high-impact TMD group only in the dense analysis, and the high-impact group exhibited the most bilateral cortical activations in both analyses. Based on anterior literature findings, we would have expected more bilateral cortical activations across all groups. A potential explanation for our observation is that increased thalamic sensitivity in high-impact TMD patients may result in augmented brain activations across bilateral cortical regions.

One of the more perplexing findings was that the control and low-impact TMD groups failed to demonstrate significant activations in the primary somatosensory cortex (SI). The role of SI in pain processing is well-documented [32,75]. Functional changes in this region are often seen in chronic pain, but findings in TMD patients have been inconsistent [77] with at least one study reporting no functional reorganization in SI [28]. In our sample, significant activations were seen in two right SI subregions in the high-impact TMD group only: 3aFace and 3aUpperExtremity.

As part of the HCP young adult neuroimaging protocol, motor contrasts involving tongue and hand movement helped define the somatotopic SI "face" and "upper limb" boundaries, respectively [78]. We used this functional localizer data from the HCP to subdivide the somatosensory cortices based on their somatotopic boundaries. The 3aFace activation found for the high-impact TMD group made theoretical sense as the noxious stimuli were applied within that body region, while activation within the 3aUpperExtremity parcel was unexpected. Previous studies in primates with amputated upper limbs [79] and spinal lesions [80] found that somatosensory stimuli to the face can evoke sensory referral to the upper extremity and can even activate SI (area 3b) in the somatotopic hand region. The original belief that these referrals marked the unmasking of latent synapses between hand and face subregions of SI was discredited by findings from one study [80], and the authors interpreted these findings to suggest that functional reorganization (e.g., axonal sprouting) following spinal cord injuries likely takes place at subthalamic levels. While the exact mechanisms underlying referred somatosensation and referred pain remain largely unknown, we speculate that our finding of

3aUpperExtremity activations in the high-impact TMD group following noxious dentoalveolar stimuli may be attributed to increased SI functional reorganization in this subgroup of patients.

Control and low-impact TMD groups failed to show significant SI activations, unlike the high-impact TMD group, in the parcellated analysis due to sub-threshold mean BOLD response magnitudes. It is worth noting that the low-impact TMD group did exhibit a large effect size in one SI face subregion (1face-right: Cohen's d = 0.89, p = 0.062). Further, significant activations in SI face subregions were seen in both TMD groups in the dense analysis, suggesting that amplified SI activations in response to noxious orofacial stimuli may be a characteristic of chronic TMD pain.

Within the posterior opercular cortices, we observed activations in parcels encompassing the classic "SII region" [81] in both TMD groups. In each TMD group, bilateral activations were seen in area OP4, while Area OP2-3 was activated in the right hemisphere of the high-impact TMD group only. Along with SI, the SII region is part of the sensory-discriminative brain network [32] which is commonly activated in experimental pain conditions [75]. A coordinate-based meta-analysis investigating brain responses to orofacial pain stimuli in healthy controls revealed consistent activations in bilateral SII [12]. Previously, when applying noxious dentoalveolar stimuli to PDAP patients and controls with matched subjective stimulus intensity, we found greater activation in right SII in PDAP patients relative to controls [39]. Cumulatively, the evidence indicates that SII subregions are commonly activated following noxious orofacial pain stimuli in both pain-free controls and those participants with chronic orofacial pain conditions. Our findings suggest that, similar to PDAP patients, chronic TMD pain patients may exhibit amplified cortical responses in these brain regions relative to controls, though these differences did not reach significance thresholds in the present study.

Other posterior opercular cortex parcels–areas 43, PFcm, and FOP1 –were activated bilaterally in the high-impact TMD group, while only the right PFcm was activated in the low-impact TMD group. Along with its role in the sensory-discriminative network, it has been proposed that this brain region is involved in the affective component of pain perception [82]. In the present study, chronic TMD pain groups scored higher than controls in nearly all questionnaires (Table 1), indicating psychosocial impairment. Moreover, scores in the high-impact TMD group tended to be higher than those in the low-impact group, though these differences were not statistically significant. We speculate that pain-related cortical activations in the posterior opercular cortex in chronic TMD pain patients, particularly those with high-impact pain, may be associated with the greater psychosocial burden present in this patient population.

In chronic TMD pain groups, we observed both activations and deactivations in brain regions involved in motivational-affective and cognitive-evaluative processes, including the subgenual anterior cingulate cortex (sgACC), insula, and orbitofrontal complex (OFC). The ACC and insula help comprise the motivational-affective cortical network, and both regions are commonly activated in response to acute pain stimuli [75,83]. Studies have also shown that the sgACC, insula, and OFC are functionally connected, and may be involved in descending pain modulation via interactions with the periaqueductal gray (PAG) [84–86].

Significant BOLD deactivations were seen in bilateral sgACC subregions in the low-impact TMD group: areas s32 (left) and 25 (right). In contrast, previous neuroimaging studies have consistently reported increased ACC activations in TMD patients [22,23]. However, most TMD neuroimaging studies have not reported deactivations when presenting their results [77], so future studies reporting the results of two-tailed statistical tests to investigate both positive and negative functional responses to sensory stimuli would help further our understanding of TMD brain mechanisms.

The only other significant BOLD deactivation in the low-impact TMD group was found in the left OFC. The OFC is involved in the voluntary control of unpleasant emotions [87] and its increased activation during mindfulness exercises and cognitive behavioral therapy suggests a role in cognitive pain modulation [88,89]. In patients suffering from migraine with aura, significantly reduced pain-evoked potential amplitudes have been shown in the OFC [90] which is in line with our present findings in TMD patients. Our results suggest that chronic TMD pain sufferers may exhibit reduced noxious stimulus-evoked OFC activity, perhaps reflective of dysfunction in descending pain modulatory pathways.

Only the high-impact TMD group exhibited significant activations within the insular and frontal opercular cortex, with multiple activated parcels bilaterally. The insula has been coined a multidimensional integration site for pain due to its role in sensory and affective processes [91]. It has been postulated that the posterior insula encodes noxious stimuli, while the anterior insula is associated with the integration of emotional and interoceptive states [68,92–94]. In our study, parcels within both the right anterior (area MI) and posterior (Pol1) insular cortices were activated in the high-impact TMD group.

The insula is also considered a key cortical region involved in allodynia, defined as pain due to a stimulus that does not normally provoke pain [95], which may contribute to the expanded pain-evoked insular activations seen across numerous chronic musculoskeletal pain conditions [96]. Previous neuroimaging studies in chronic TMD pain patients have revealed metabolic alterations in the right posterior insula [31], increased gray matter volume (GMV) in the right anterior insula [20], as well as increased functional connectivity between the right anterior insula and right thalamus [22]. In our previous study we showed that PDAP patients exhibited augmented pain-evoked insular activations compared to controls in response to dentoalveolar stimuli, both when matched to stimulus intensity and to subjective pain ratings [39]. Despite the absence of between-group differences, the mean group activations found in the present study suggest that pain-evoked brain responses in anterior and posterior insular cortices may be augmented in high-impact TMD pain patients.

Temporo-parieto-occipital junction (TPOJ) activations were seen in both TMD groups. This region is involved in multiple high-level functions including language, vision, reading, and memory [97]. In the past decade, evidence has emerged that the TPOJ is involved in processing salient sensory stimuli, including pain [32]. One study demonstrated a rightward asymmetry in white matter connectivity between the TPOJ and insula in healthy subjects, supporting a role for these regions in stimulus-driven attention and pain [98]. Area TPOJ1 was activated in the right hemisphere of the high-impact TMD group, while peri-sylvian language area (PSL) activations were seen bilaterally in the high-impact TMD group and in the right hemisphere of the low-impact TMD group. Area PSL has strong functional connectivity with area 55b [78], the premotor cortex parcel activated in the right hemisphere of the high-impact TMD group.

Language area activations like area PSL are not commonly reported in the pain neuroimaging literature, but premotor cortex changes have been seen in patients with chronic orofacial pain. One study found that chronic TMD pain patients showed increased cerebral blood flow in the premotor cortex compared to controls [99], which the authors theorized was due to the sensitized masticatory muscle nociceptors in these patients. Augmented premotor cortex activations were also noted in PDAP patients compared with controls following painful dentoalveolar stimuli [39]. These collective findings suggest that functional changes in the premotor cortex may exist both at rest and in response to orofacial noxious stimuli in chronic TMD pain patients.

## Conclusions

Brain activations evoked by dentoalveolar noxious pressure stimulation were found to be amplified in high-impact TMD pain, however its magnitude was not large enough to result in significant between-group differences given the limited sample size of the present study. Numerous pain-related subcortical regions were exclusively activated in the high-impact TMD group, including the nucleus accumbens, putamen, caudate, and thalamus. Using a modified multimodal cortical and subcortical parcellation scheme for the stimfMRI analysis, a gradient of parcels surviving thresholding was seen according to pain-impact status, with the high-impact TMD group exhibiting the most activations relative to the low-impact TMD and pain-free control groups. These brain activations occurred in cortical regions associated with sensory-discriminative and motivational-affective pain processing, as well as other higher-order functions including language and cognition.

Based on our findings, future studies should attempt to characterize the roles of specific brain regions in the expression of chronic TMD pain. Our study suggests that insular activations to nociceptive stimuli may be amplified in chronic TMD pain patients with higher pain-impact status. Investigation of activations in insular parcels across patients with different pain-impact levels could clarify the role of this important region in chronic TMD pain patients. Further investigation to corroborate our findings in other regions, such as deactivations seen in the sgACC and OFC, will better our understanding of brain mechanisms related to chronic pain in this patient population.

Our results suggest that functional brain abnormalities related to sensory processing are potentially involved in chronic TMD pain. We applied noxious stimuli to a relatively small body area in a site not typically associated with clinical TMD presentation, yet high-impact TMD participants processed these stimuli using brain resources in a more widespread manner based on their brain activations. Further, we investigated TMD pain-impact status and brain functioning using relatively novel and advanced neuroimaging data acquisition and analysis techniques developed by the HCP consortium. Future neuroimaging studies using the multimodal parcellation and implementing different experimental interventions and improved methodology (e.g., larger sample size, equal left-right stimulus distribution, use of two-tailed statistical testing, effect size reporting) will help better our understanding of brain mechanisms involved in chronic TMD pain and potentially support the development of targeted therapies and improved prognosis and for this patient population.

## Acknowledgments

The authors wish to thank Ms. Patt Carlson for her efforts in screening potential study participants. We would also like to acknowledge the following for their significant roles in in the earlier stages of this research project: Alberto Herrero Babiloni DDS, MS (McGill University) and Flavia P. Kapos DDS, MS (University of Washington). The authors acknowledge the Minnesota Supercomputing Institute (MSI) at the University of Minnesota for providing resources that contributed to the research results reported within this paper. URL: http://www.msi.umn.edu.

## Author Contributions

**Conceptualization:** Connor M. Peck, David A. Bereiter, Christophe Lenglet, Estephan J. Moana-Filho.

**Data curation:** Estephan J. Moana-Filho.

**Formal analysis:** Connor M. Peck, Lynn E. Eberly, Estephan J. Moana-Filho.

**Funding acquisition:** Estephan J. Moana-Filho.

**Investigation:** Connor M. Peck, Estephan J. Moana-Filho.

**Methodology:** Estephan J. Moana-Filho.

**Project administration:** Connor M. Peck, Estephan J. Moana-Filho.

**Resources:** Connor M. Peck, Estephan J. Moana-Filho.

**Software:** Estephan J. Moana-Filho.

**Supervision:** Estephan J. Moana-Filho.

**Visualization:** Connor M. Peck, Estephan J. Moana-Filho.

**Writing – original draft:** Connor M. Peck.

**Writing – review & editing:** David A. Bereiter, Lynn E. Eberly, Christophe Lenglet, Estephan J. Moana-Filho.

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
