## [Decision Letter · Decision Letter 0]

10 Jul 2022

PONE-D-22-08130Altered Brain Responses to Noxious Dentoalveolar Stimuli in High-Impact Temporomandibular Disorder Pain PatientsPLOS ONE

Dear Dr. Moana-Filho,

Thank you for submitting your manuscript to PLOS ONE. After careful consideration, we feel that it has merit but does not fully meet PLOS ONE’s publication criteria as it currently stands. Therefore, we invite you to submit a revised version of the manuscript that addresses the points raised during the review process. When you read the comments of the reviewers, you will see that reviewer 1 suggests dividing the manuscript in two or more separate studies.However, in my opinion it is not necessary to divide the manuscript, and therefore this issue is not crucial for acceptance.

We look forward to receiving your revised manuscript.

Kind regards,

Peter Schwenkreis

Academic Editor

PLOS ONE

Journal Requirements:

Research reported in this publication was supported by the National Institute of Dental & Craniofacial Research of the National Institutes of Health under Award Number R00 DE027414. 

However, funding information should not appear in the Acknowledgments section or other areas of your manuscript. We will only publish funding information present in the Funding Statement section of the online submission form. 

Research reported in this publication was supported by the National Institute of Dental & Craniofacial Research of the National Institutes of Health (https://www.nidcr.nih.gov) under Award Number R00 DE027414 (E.J.M.).  The funder had no role in study design, data collection and analysis, decision to publish, or preparation of the manuscript.

4. We note that you have included the phrase “data not reported” in your manuscript. Unfortunately, this does not meet our data sharing requirements. PLOS does not permit references to inaccessible data. We require that authors provide all relevant data within the paper, Supporting Information files, or in an acceptable, public repository. Please add a citation to support this phrase or upload the data that corresponds with these findings to a stable repository (such as Figshare or Dryad) and provide and URLs, DOIs, or accession numbers that may be used to access these data. Or, if the data are not a core part of the research being presented in your study, we ask that you remove the phrase that refers to these data.

Reviewers' comments:

Reviewer's Responses to Questions

**Comments to the Author**

1. Is the manuscript technically sound, and do the data support the conclusions?

Reviewer #1: Partly

Reviewer #2: Yes

2. Has the statistical analysis been performed appropriately and rigorously? 

Reviewer #1: Yes

Reviewer #2: Yes

3. Have the authors made all data underlying the findings in their manuscript fully available?

Reviewer #1: Yes

Reviewer #2: Yes

4. Is the manuscript presented in an intelligible fashion and written in standard English?

Reviewer #1: Yes

Reviewer #2: Yes

5. Review Comments to the Author

Reviewer #1: Please see the attached file.

Reviewer #2: This is a concise and interesting report describing the brain responses for patients with chronic TMD pain. In my opinion the paper appears acceptable for publication but there are a few revisions. Please check as below.

１）Is maladapted brain plasticity of chronic fascial pain possible due to depression or anxiety?

２）The reflexive reaction seems to be getting bigger, but what about it?

6. PLOS authors have the option to publish the peer review history of their article (what does this mean?). If published, this will include your full peer review and any attached files.

Reviewer #1: No

Reviewer #2: No

---

## [Author Response · Author response to Decision Letter 0]

23 Aug 2022

Title: Altered Brain Responses to Noxious Dentoalveolar Stimuli in High-Impact Temporomandibular Disorder Pain Patients

Authors: Connor Peck, DDS, MS; David A. Bereiter, PhD; Lynn E. Eberly, PhD; Christophe Lenglet, PhD; Estephan Moana-Filho, DDS, MS, PhD

Reviewer’s comments and author’s reply

Date of review: July 10, 2022

Dear Dr. Peter Schwenkreis

We would like to thank you for considering our manuscript for publication in your journal and also thank the reviewers who took the time to review and make pertinent comments aimed at improving its quality.

We addressed each of the issues raised by the reviewers, and incorporated modifications to the resubmitted manuscript where appropriate. Please note that in order to address some of the comments, more text was incorporated.

Below, we have copied the reviewers’ comments in black and our detailed responses to each point are written in red.

Reviewer #1

The manuscript by Peck et al. aims to determine brain activations in response to noxious dentoalveolar pressure stimuli in chronic TMD patients as well as the CNS effects of pain impact status. I would certainly like to commend the authors for a very thorough study and well-written manuscript. The best available methods have been used and a good and well-defined population was included. The findings in the present manuscript will enable important future specific studies on the mechanisms that are proposed in the findings. 

I have only a few major and a few minor comments, see below.

MAJOR POINTS

This is a very, very extensive study. I mean that both from a positive and a challenging aspect. In my mind, this is too much to put into a single manuscript. I would urge the authors to consider dividing the manuscript into two (or more?) separate studies in order to improve the possibilities for the readers to understand all your interesting findings. 

Authors’ response: We understand this feedback and agree that the neuroimaging data acquired in our study is robust. Ultimately, our goal was specifically to compare the brain activations across three groups of participants: 1. Pain-free controls; 2. Chronic TMD low-impact pain; 3. Chronic TMD high-impact pain. In the results, we highlighted each of the brain regions activated across groups using two different neuroimaging analysis approaches. Respectfully, it is our opinion that dividing the current manuscript into multiple manuscripts would hinder the clarity for the group comparisons report that is central to this manuscript.

The manuscript, in its present form, is more of an hypothesis-generating type than an hypothesis-testing type. Nothing wrong with that, especially regarding brain activation by nociceptive stimuli, but I would suggest to change the approach to an hypothesis-generating type of study. And for the manuscript to suggest hypotheses worth specifically investigating in future studies.

Authors’ response: We agree that this is a hypothesis-generating type of study. We selected this approach for a few reasons. As stated in the Introduction, previous studies involving application of noxious stimuli to distant body sites in TMD patients are limited and results are heterogenous. Moreover, this was the first neuroimaging study to subdivide chronic TMD pain participants based on pain-impact status, so it was unclear if, and how, this may affect our results. Because of the limited data in the literature related to our topic, we decided to take the hypothesis-generating approach. 

The recommendation that we suggest hypotheses to investigate in future studies was noted and we agreed. We added a paragraph in the Conclusions section (pg. 35) to accomplish satisfy this recommendation.

There were no significant between-group findings (although interesting and important within-group findings). The authors should discuss this matter more in detail, separately. In the present manuscript a very short sentence is placed in the Conclusion. But what does this mean? Doesn’t chronic TMD pain elicit more brain activation from this stimuli than in controls, i.e. is chronic TMD pain not related to the activation caused by this acute form of stimuli?

Authors’ response: We felt we had expanded the discussion about these findings in the first three paragraphs of the Discussion section (pgs. 26-27). Based on the findings, there certainly appears to be a trend showing that chronic TMD pain patients display greater brain activations in response to our stimulus than controls, but these between-group difference did not reach statistical significance, at least partially due to the factors we outlined.  

The Conclusion is a repetition of the result more than a generalizing conclusion based on the findings. Please rewrite with point no 2 in mind. 

Authors’ response: We attempted to offer a general conclusion in the first two sentences of the third paragraph in the Conclusion section (pg. 35): 

“Our results suggest that functional brain abnormalities related to sensory processing are potentially involved in chronic TMD pain. We applied noxious stimuli to a relatively small body area in a site not typically associated with clinical TMD presentation, yet high-impact TMD participants processed these stimuli using brain resources in a more widespread manner judging by their brain activations.”

We hope that the addition of the second paragraph (pg. 35) adds another layer to this section and outlines that based on our results, more research is still needed to improve our understanding of chronic TMD pain neural processes.

The Figure legends must be in much more detail  

Authors’ response: We tried to comply to the reviewer’s comment to the best of our perspective, given the lack of information of what figures needed more details. We added detail to the caption for Fig 1. We added a descriptive sentence for Fig 4. We added more information in the captions for Fig 5 and 6 for clarification. Finally, we felt that no further information was needed for Figs 2 and 3.

MINOR POINTS

Row 78 Please consider changing ”central sensitisation” to ”maladaptive central pain modulation” or something similar. Central sensitisation is one (of several) specific mechanism whereas maladaptive central pain modulation is a better descriptive term collectively describing all the changes in central pain processing in chronic pain (sensitisation, desensitisation, somatosensory changes, autonomi changes etc).

Authors’ response: We agree with the reviewer’s suggestion. We added the term maladaptive central pain modulation and used it throughout the manuscript in place of central sensitization (pgs. ii, 1, 2, 28).

Row 138 Provide reference to the IRB-approved script

Authors’ response: The IRB-approved script is simply an operationalization of the study inclusion and exclusion criteria in the Methods section, “Participants” subsection, where we offer relevant references for those criteria. This way, the references relevant for the inclusion and exclusion criteria also apply to the IRB-approved script.

Row 172- Provide references to all included instruments

Author’s response: This comment was unclear to the authors. In the methods section, “Informed Consent, TMD Examination & Questionnaires (Visit 1)” subsection, we offered several relevant references for all instruments used. For the DC/TMD Axis II instrument, we included the reference for the DC/TMD main article by Schiffman et al., 2014. We believe that this should suffice for the interested readers to find all the pertinent scientific background literature, while at the same time we prevent inflating the number of references included in our manuscript.

Row 197 & 229 Please provide data for the intervals between the sessions

Authors’ response: In the first paragraph of the Methods section (pg. 3), we stated that the time between visits ranged from 2-7 days. However, we believe that the study visit of interest for the data and results reported in this manuscript is visit 3, when the neuroimaging session took place. We added a paragraph to the Results section (pg. 17) detailing the intervals between visits 2 and 3 for each group.

Row 208 If possible, please provide a figure

Authors’ response: Our group has published two articles on the development and use of the dentoalveolar stimulus device, which contains several pictures and its technical description. We included the references to the two articles where relevant, which should help the interested reader to look into more details about the device while avoiding cluttering the present manuscript, given the considerable amount of information about the device already included throughout the manuscript.

Row 358- As I understand, degree of depression and/or anxiety was not controlled for in the MR analyses. This is common in MR analyses since the known effect of depression/anxiety on brain activation. Please motivate what this was not performed or (maybe better?) recalculate. It is not impossible that the possibilities to find significant between-group differences will increase.

 Authors’ response: Thank you for this comment. Indeed, we did this analysis by adding the mean scores for PHQ-9 (depression) and GAD-7 (anxiety) among others as covariates in the general linear models tested for the fMRI analysis, but no statistically significant results were found. We added a paragraph stating this exploratory analysis at the end of the methods section (pg. 16) and also in the results (pg. 26).

Point 2.4.2 Commends for the extensive and top-notch statistics used! 

Authors’ response: Thank you! We really appreciate the great feedback you provided.

Rows 452-482 Belongs to M&M (if not stated otherwise in the instructions)

Authors’ response: We provided a detailed account of the participant characteristics in the first few paragraphs of the "Results” section. Several articles in the published literature vary in how they report their sample characteristics, in either the “Methods” or “Results” section. In our case, we believe that placing this information in the “Results” section of our manuscript benefits the reader given the flow of information that follows it (Pain sensitivity confounders, dentoalveolar pressure pain ratings).

Reviewer #2

This is a concise and interesting report describing the brain responses for patients with chronic TMD pain. In my opinion the paper appears acceptable for publication but there are a few revisions. Please check as below.

１）Is maladapted brain plasticity of chronic fascial pain possible due to depression or anxiety?

Authors’ response: It is known that chronic pain patients generally exhibit higher depression and anxiety scores, and our results from the GAD-7 and PHQ-9 found this to be the case. We ran a separate fMRI analysis controlling for depression and anxiety scores which did not show statistically significant results, so it did not appear to play a role in our findings as described in the response to the comment above.

２）The reflexive reaction seems to be getting bigger, but what about it?

Authors’ response: It is not clear to us what the reviewer meant by “reflexive reaction”. Assuming it refers to the wider cortical activation following painful dentoalveolar pressure stimulation in the chronic TMD high-impact group, we speculate that this finding suggests presence of pain amplification in this group of TMD cases. The specific brain mechanisms related to such pain amplification are yet to be determined and based on anterior literature we argue that the presence of maladaptive central pain modulation mechanisms in chronic TMD high-impact pain could at least partially explain this finding.

---

## [Editor Report · Decision Letter 1]

26 Aug 2022

Altered Brain Responses to Noxious Dentoalveolar Stimuli in High-Impact Temporomandibular Disorder Pain Patients

PONE-D-22-08130R1

Dear Dr. Moana-Filho,

We’re pleased to inform you that your manuscript has been judged scientifically suitable for publication and will be formally accepted for publication once it meets all outstanding technical requirements.

Kind regards,

Peter Schwenkreis

Academic Editor

PLOS ONE
---

## [Editor Report · Acceptance letter]

19 Sep 2022

PONE-D-22-08130R1 

Altered Brain Responses to Noxious Dentoalveolar Stimuli in High-Impact Temporomandibular Disorder Pain Patients 

Dear Dr. Moana-Filho:

I'm pleased to inform you that your manuscript has been deemed suitable for publication in PLOS ONE. Congratulations! Your manuscript is now with our production department. 

Kind regards, 

on behalf of

Dr. Peter Schwenkreis 

Academic Editor

PLOS ONE